# Weakly-Supervised Multi-Granularity Map Learning for Vision-and-Language Navigation

**Peihao Chen**[1,6*]   **Dongyu Ji**[1*]   **Kunyang Lin**[1,2]   **Runhao Zeng**[5]
**Thomas H. Li**[6]   **Mingkui Tan**[1,7†]   **Chuang Gan**[3,4]
[1]South China University of Technology, [2]Pazhou Laboratory,
[3]MIT-IBM Watson AI Lab, [4]UMass Amherst, [5]Shenzhen University,
[6]Information Technology R&D Innovation Center of Peking University,
[7]Key Laboratory of Big Data and Intelligent Robot, Ministry of Education,
{phchencs, dongyuji.jdy}@gmail.com, mingkuitan@scut.edu.cn

## Abstract

We address a practical yet challenging problem of training robot agents to navigate in an environment following a path described by some language instructions. The instructions often contain descriptions of objects in the environment. To achieve accurate and efficient navigation, it is critical to build a map that accurately represents both spatial location and the semantic information of the environment objects. However, enabling a robot to build a map that well represents the environment is extremely challenging as the environment often involves diverse objects with various attributes. In this paper, we propose a multi-granularity map, which contains both object fine-grained details (*e.g.*, color, texture) and semantic classes, to represent objects more comprehensively. Moreover, we propose a weakly-supervised auxiliary task, which requires the agent to localize instruction-relevant objects on the map. Through this task, the agent not only learns to localize the instruction-relevant objects for navigation but also is encouraged to learn a better map representation that reveals object information. We then feed the learned map and instruction to a waypoint predictor to determine the next navigation goal. Experimental results show our method outperforms the state-of-the-art by 4.0% and 4.6% *w.r.t.* success rate both in seen and unseen environments, respectively on VLN-CE dataset. Code is available at https://github.com/PeihaoChen/WS-MGMap.

## 1 Introduction

Developing a robot that is able to cooperate with humans is one of the goals for embodied artificial intelligence. An important ability for such a robot is to understand human instructions (*e.g.*, "*stop between the gray sofa and wooden table*") and navigate to the corresponding location. Toward this goal, Anderson *et al.* [1] propose a vision-and-language navigation (VLN) task, which requires an agent to follow human language instructions to navigate in unseen environments. In this paper, we focus on its variant task VLN-CE [32], where the agent navigates in a continuous environment. The agent could only perform low-level actions and perceive RGB-D with a limited field of view.

Current dominant methods for this task are designed in an end-to-end manner [26, 32, 40]. They perceive the environment implicitly from raw RGB-D images and predict a sequence of actions using a recurrent model. These methods attempt to learn mapping, instruction-vision correspondence, and path planning implicitly, which increases the learning difficulty. To reduce the learning difficulty,

---

*Equal contribution.
†Corresponding author.

36th Conference on Neural Information Processing Systems (NeurIPS 2022).

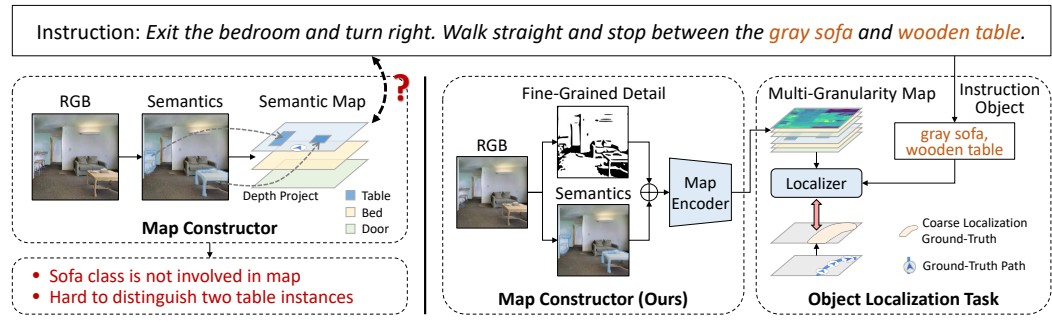

Figure 1: Existing semantic map (a) can only represent a part of environment object classes without attribute details. Our multi-granularity map (b) contains extra fine-grained environment details (*e.g.*, texture, color) and learns to represent diverse objects with detailed attributes through a weakly-supervised object localization task.

modular map-based approaches [6, 19, 27] equip the embodied agent with a semantic map to perceive the environment. This map is built by projecting RGB image segmentation results to an egocentric top-down map (shown in Figure 1 (a)), and reveals the location and semantic class of a part of the environment objects.

However, this map is hard to represent all object classes (*e.g.*, sofa is not represented in Figure 1 (a)). More critically, it can not represent the attributes of objects (*e.g.*, shape, color, texture and material), which are critical for localizing an object. As a result, it provides insufficient information to discriminate the "*long wooden dining table*" from two table instances shown in Figure 1 (a). We find that such ambiguity cases appear frequently in an indoor environment. Detailed statistical data and visualization examples are shown in Appendix.

In this paper, we propose to learn a map that represents the semantics and attributes of diverse objects in a weakly-supervised manner. We achieve this goal in two steps. First, we augment the existing semantic map to a multi-granularity map as shown in Figure 1 (b). This map contains different granularity information, namely high-level object semantics recognized by a segmentation model, and low-level object fine-grained details. To get the low-level details, we follow existing works [4] to project high-dimensional segmentation model latent features to a top-down map. These latent features have proven to contain rich object details such as color, texture, and shape [28, 52].

Second, to make the map better represent instruction objects using the multi-granularity features described above, we propose an instruction-relevant object localization auxiliary task as shown in Figure 1 (b). Specifically, we feed the map representation and instruction objects to a localizer to predict the location of these objects. Instead of manually annotating the object location on a map as localization ground-truth, we automatically generate a coarse localization ground-truth from instruction-path paired data, considering map regions that are close to the instruction path as coarse ground-truth. From this task, agents learn to localize instruction-relevant objects for finding an instruction-relevant path. More critically, to localize instruction objects, the map encoder is encouraged to reason a map representation that reveals the precise semantics of each map region.

We then feed the learned map and the instruction to a waypoint predictor to determine the next navigation goal. An existing off-the-shelf local policy [47] is used to determine a low-level action (*e.g.*, go forward, turn left or right) to go for the navigation goal. We name our method as weakly-supervised multi-granularity map (**WS-MGMap**) for VLN. Experimental results on VLN-CE benchmark dataset show our proposed method outperforms state-of-the-art methods.

To sum up, our main contributions are as follows: 1) We construct a multi-granularity map to represent both fine-grained details and abstract semantics information of the environment. To our best knowledge, it is the first time to introduce multi-granularity knowledge in a map format for VLN task. 2) We propose a weakly-supervised object localization auxiliary task, from which agents learn to leverage multi-granularity information to infer a discriminative map representation without the need for manual map annotation. 3) With WS-MGMap, our agent robustly localizes objects that are incorrectly recognized by segmentation models. On VLN-CE [32], our method improves navigation success rate by 4.0% and 4.6% in seen and unseen environments, respectively.

## 2   Related Works

**Vision-and-language navigation.**   VLN task [2] has drawn significant attention in embodied AI domain. Early works focus on data augmentation methods [13, 44], auxiliary tasks [35, 46, 54] and pre-training methods [21, 23, 36] on the discrete environments in which the agents can only perceive from a sparse set of points. However, these methods can not perform well when the agent navigates in continuous 3D environments [25, 15]. To address this issue, Krantz *et al.* [32] exploits Habitat simulator [42] to convert discrete trajectory paths to continuous trajectories and proposes a continuous VLN setting. Under these settings, Krantz *et al.* [32] and Raychaudhuri *et al.* [40] propose the end-to-end method in which the agent takes a representation of the visual observation and instructions as input at each time step to predict an action. Another line of works [19, 31] tackles the VLN problems by leveraging the language-conditioned waypoint. Chen *et al.* [8], Georgakis *et al.* [19] and Irshad *et al.* [27] build a structured semantic top-down map, which is more relevant to our work. However, those map-based methods are limited in representing all object classes or the attributes of objects, which prevents the agent from effectively grounding instructions in maps. In contrast to these works, our method learns a map that represents the semantics and attributes of diverse objects in a weakly-supervised manner.

**Map representation for navigation.**   Perceiving the environment through a map representation is helpful for navigation tasks. Previous works propose different methods for building a map, each of them focusing on representing different information on the map. For example, occupancy map indicates whether a point is occupied and provides location information for point-goal navigation and exploration [29, 30, 39, 9, 17, 16, 14]. Topological maps encode the relationship information among different nodes in the environment and have been explored to tackle different navigation tasks [3, 7, 8, 37]. Semantic maps represent the semantic information of the environment and has been successfully applied in object-goal navigation [6, 10, 50]. Concurrent works [19, 27] attempt to leverage semantic map for VLN task. Existing works [24, 4, 22, 11, 20] try to generate a deep feature map by projecting high-dimensional features encoded from neural network to a top-down map, which is used for localizing robot [24], predicting semantic map [4], navigating to a few target objects [22, 20], and 3D reconstructing objects [11]. However, as compared to object-goal navigation, the instructions in VLN involve more object information. It is challenging to ground rich object information in the map described above individually due to their respective limitations. Unlike these maps, we propose a multi-granularity map that contains complementary granularity information for representing the environment more comprehensively.

**Weakly-supervised learning for object localization.**   Weakly-supervised object localization is a challenging task that requires learning to localize objects given solely category information in both image[55, 34, 53, 51, 48] and video[33, 49]. CAM[53, 51, 48] has been widely used in weakly-supervised object localization by learning a heat map representing the potential of object location. In contrast to this literature, we aim to utilize the weakly-supervised object localization task to learn discriminative map representation that provides environment information for VLN by localizing instruction-relevant objects.

## 3   Vision-and-Language Navigation using Multi-Granularity Map

### 3.1   Problem formulation

We consider the vision-and-language navigation task in a continuous environment [32] (VLN-CE), where an agent is required to follow a specific path described by the natural language instruction $I$. Compared with the original VLN task, in VLN-CE settings, agents can not access to predefined navigable graph, so the agent is required to perform navigation through a sequence of low-level actions $a \in \mathcal{A} = \{\text{FORWARD}, \text{TURN-LEFT}, \text{TURN-RIGHT} \text{ and } \text{STOP}\}$. We equip the agent with an RGB-D camera with a limited field of view, capturing a $224 \times 224$ RGB image $R$ and depth image $D$ at each time step. Although some current works use predefined navigable graph [8], continuous-space actions [31] and panoramic cameras [13, 25] to improve the performance, we strictly keep the same settings with VLN-CE [32], which are closer to reality and more challenging.

Existing VLN methods [31, 32, 40] attempt to perceive the environment implicitly from RGB-D observation, which often requires a large number of training data. In contrast, humans will

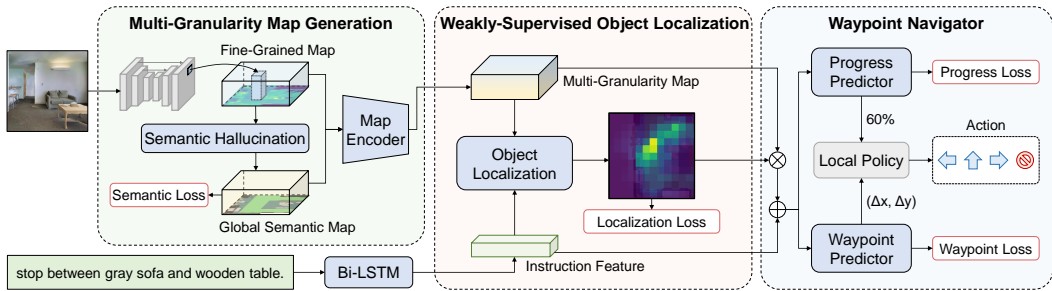

Figure 2: General scheme of WS-MGMap for VLN task. We assemble both fine-grained details and semantic information about environments to build a multi-granularity map. Agents learn to leverage such information for representing diverse objects through a weakly-supervised object localization task. The learned map and instruction are then fed to a waypoint navigator for deciding actions.

build a map-like environment representation explicitly to provide necessary information for VLN. Equipping an agent with the mapping ability like humans is challenging because it is hard to represent diverse environment objects and their attributes in a map. In this paper, we solve this challenge from two aspects: 1) we augment a semantic map with extra fine-grained environment details (*e.g.*, texture, color), which are extracted from latent features of a pre-trained segmentation model, to build a multi-granularity map. These different granularity features provide necessary information for representing diverse objects and attributes. 2) To make the multi-granularity map representation more discriminative, we propose a weakly-supervised localization auxiliary task for learning correspondence between map representation and instruction objects. With the learned multi-granularity map, we feed it and the instruction to a waypoint navigator to decide the next navigation action. The general scheme is shown in Figure 2.

## 3.2 Perceiving environment via multi-granularity map

To better represent environment objects on a map, we propose to leverage both fine-grained features and semantic features about objects for building a multi-granularity map. We next introduce how to capture these two types of features for map representation.

**Fine-grained map.** To capture fine-grained environment details, we feed RGB images captured at each time step to a pre-trained segmentation model (a U-Net [41] in our case). Previous network interpretability works [28, 52] show that latent features from different layers of this model contain different types of fine-grained details of objects, *i.e.*, low-level features represent color and texture while high-level features represent object parts. Based on this observation, we use these latent features to represent environment details and project the latent features to an egocentric top-down map [4]. We name the projected map as fine-grained map $\mathbf{M}^f \in \mathbb{R}^{m \times m \times c_f}$, where $m \times m$ is map size and $c_f = 64$ is the number of channels of the projected latent features. Each map pixel represents $12cm \times 12cm$ space of environments. We choose to project latent features from the last layer of U-Net, which contains both low-level and high-level features because of the skip connection mechanism in U-Net. We also evaluate the effect of other layer features in experiments.

**Global semantic map.** The fine-grained map aggregates environment details from RGB images which are captured along the navigation path. Because camera has a limited field of view, the fine-grained map does not contain environmental information that has not been observed. More importantly, requiring agents to recognize all objects, including some common-seen objects, from fine-grained details increases learning difficulty. Thus, to hallucinate environments beyond the field of view and to get semantic information about common-seen objects, we feed the fine-grained map to a semantic hallucination module to predict a global semantic map $\mathbf{M}^s \in \mathbb{R}^{m \times m \times c_s}$, where each pixel indicates whether a particular object out of $c_s = 27$ object categories locates in the corresponding space. The $c_s$ object categories are listed in Appendix. The hallucination module consists of three convolution layers with batch norm and ReLU activations, followed by a U-Net which has two encoder and two decoder convolutional blocks with skip connections. We get the ground-truth global semantic map $\bar{\mathbf{M}}^s$ from the available 3D semantic information in Matterport3D following existing

methods [18, 19]. A pixel-wise semantic loss $l_s$ is defined as follows:

$$l_s = \sum_i \text{CrossEntropy}(\mathbf{M}_i^s, \bar{\mathbf{M}}_i^s). \tag{1}$$

**Multi-granularity map.** We merge information from both the fine-grained map and predicted semantic map using a map encoder. Specifically, these two maps are first processed by a convolution layer, respectively. The concatenation of two processed maps is fed to a convolution layer for producing a multi-granularity map $\mathbf{M} \in \mathbb{R}^{m \times m \times c}$, where $c$ is the number of map channels.

## 3.3 Weakly-supervised map representation learning via object localization task

To learn the correspondence between multi-granularity map representation and diverse objects, we propose a weakly-supervised instruction-relevant object localization auxiliary task. To finish this task, the agent is required to reason precisely about semantics information for each region of map representation, which makes the map representation more significant.

**Formulation of weakly-supervised auxiliary task.** To localize the instruction object on a map, we feed the instruction and multi-granularity map to an object localization module to predict an egocentric grid map $\hat{\mathbf{P}} \in \mathbb{R}^{m \times m}$ which indicates a 2D distribution of potential locations of instruction-relevant objects. Because we do not know the exact location of these instruction objects, we use the instruction-relevant path as guidance to generate a coarse ground-truth egocentric grid map $\mathbf{P} \in \mathbb{R}^{m \times m}$. The principle for generating coarse ground-truth is that the regions closer to the path have a higher probability of containing instruction-relevant objects. Specifically, we first calculate euclidean distance $d_k$ from the $k$-th map region to the path. A normalized distance map is defined as $\mathbf{P}' = \{\frac{d_{\max} - d_k}{d_{\max} - d_{\min}}\}_{k=1}^{m \times m}$, where $d_{\max}$ and $d_{\min}$ are the maximum and minimum distances among $m \times m$ regions, respectively. For a region traversed by the path, $d_k = d_{\min} = 0$ and $\mathbf{P}'_k = 1$; for the farthest region, $d_k = d_{\max}$ and $\mathbf{P}'_k = 0$. The coarse ground-truth grid map $\mathbf{P}$ is the softmax across all grids in the distance map, *i.e.*, $\mathbf{P} = \text{Softmax}(\mathbf{P}')$. An object localization loss $l_o$ is defined as KL divergence between predicted and coarse ground-truth distributions, *i.e.*,

$$l_o = \sum_{i,j} \mathbf{P}_{i,j} \log \frac{\mathbf{P}_{i,j}}{\hat{\mathbf{P}}_{i,j}}. \tag{2}$$

By minimizing the above loss function, the agent is encouraged to learn a better map representation for localizing instruction objects.

**Instruction-relevant object localization module.** The localization module contains two main components, namely a state encoder and a state-instruction attention $Att(\cdot)$. The state encoder is a gated recurrent unit (GRU) [12] and attention module is a scaled dot-product attention [45]. Specifically, we first track visual history, which consists of RGB image $R$, depth image $D$, and multi-granularity map $\mathbf{M}$, using the state encoder to encode current episode state $\mathbf{s}_t$. Then, the state $\mathbf{s}_t$ attends to instruction features using the state-instruction attention to generate attended instruction feature $\bar{\mathbf{i}}$, where the instruction features are encoded by a bi-directional LSTM [43] from instruction $I$. With the attended instruction feature $\bar{\mathbf{i}} \in \mathbb{R}^c$ and a multi-granularity map representation $\mathbf{M} \in \mathbb{R}^{m \times m \times c}$, we predict a localization grid map $\hat{\mathbf{P}} \in \mathbb{R}^{m \times m}$ by calculating the cosine similarity between $\bar{\mathbf{i}}$ and each map region feature in multi-granularity map:

$$\hat{\mathbf{P}} = \text{Softmax}((\mathbf{W}_q \bar{\mathbf{i}})(\mathbf{W}_k \mathbf{M})^T), \tag{3}$$

$$\text{where } \bar{\mathbf{i}} = \text{Att}(\mathbf{s}_t, \text{BiLSTM}(I)), \quad \mathbf{s}_t, \mathbf{h}_t = \text{GRU}([\mathbf{M}, f_R(R), f_D(D)], \mathbf{h}_{t-1}),$$

$\mathbf{W}_q, \mathbf{W}_k$ are learnable parameter matrices, $\mathbf{h}_t$ is hidden state of GRU at time step $t$, $[\cdot, \cdot]$ indicates concatenation operation, $f_R(\cdot)$ and $f_D(\cdot)$ are off-the-shelf RGB and depth encoders, respectively.

## 3.4 Waypoint navigator and overall learning objective

With the learned multi-granularity map, we feed it together with the instruction to a waypoint navigator for predicting navigation action. Specifically, an attended map features $\bar{\mathbf{m}}$ is produced by weighted averaging multi-granularity map $\mathbf{M}$ using predicted localization results $\hat{\mathbf{P}}$ in the auxiliary task described in Section 3.3. The localization results help agents highlight relevant regions that

possibly contain instruction-relevant objects. Then, following CMA [32], another state encoder (GRU) is exploited to predict a state $s'_t$ from the concatenation of attended map features $\bar{\mathbf{m}}$, attended instruction features $\bar{\mathbf{i}}$ and the first state feature $\mathbf{s}_t$, which can be formulated as follows:

$$\mathbf{s}'_t, \mathbf{h}'_t = \text{GRU}([\bar{\mathbf{m}}, \bar{\mathbf{i}}, \mathbf{s}_t], \mathbf{h}'_{t-1}), \tag{4}$$

where $\mathbf{h}'$ is a hidden state of GRU. We feed the state $s'_t$ to a waypoint predictor to predict a waypoint $\hat{\mathbf{w}} = (\Delta x, \Delta y)$ indicating the next navigation goal. We also feed the state to a progress predictor to predict a progress value $\hat{p}$ indicating the completeness of navigation process. Both of these predictors are a fully-connected layer. To get the ground-truth waypoint, we first plan the shortest path from the current agent position to the nearest waypoint on the instruction-relevant path following LAW [40]. Then, we draw a circle with 3 meters radius centered at the agent position. The intersection point between the path and circle is considered a ground-truth waypoint $\mathbf{w}$.

The ground-truth progress is defined as the normalized distance from the agent current position to the goal following existing work [32]. The waypoint loss $l_w$ and progress loss $l_p$ are defined as follows:

$$l_w = ||\hat{\mathbf{w}} - \mathbf{w}||^2, \quad l_p = ||\hat{p} - p||^2. \tag{5}$$

With a predicted waypoint, we feed it to an off-the-shelf deep reinforcement learning model DD-PPO [47] to determine a low-level action from action space $\mathcal{A}$. Note that the DD-PPO model does not update during training. If the predicted progress is higher than a threshold $\lambda_p$, the agent executes STOP action. We update the waypoint every 3 time steps.

The overall learning objective for our method is as follows:

$$L = l_s + \alpha l_o + \beta l_p + \gamma l_w, \tag{6}$$

where $\alpha$, $\beta$ and $\gamma$ are hyper-parameters.

## 4 Experiments

### 4.1 Experimental setups

**Dataset and evaluation metrics.** We conduct our experiments on VLN-CE dataset, which contains 16,844 path-instruction pairs from 90 scenes in Matterport3D. It also contains about 150k augmented data generated by EnvDrop [44]. The dataset is split into the train, seen validation, unseen validation, and test set. We follow the existing works [19, 32] to evaluate the navigation performance using success rate (SR), oracle success rate (OS), success weighted by path length (SPL), trajectory length (TL), and navigation error from goal (NE). Note that an episode is considered successful if the agent calls STOP action within 3m of the goal. More details about evaluation metrics are put in Appendix.

**Implementation details.** We implement our method based on Pytorch framework [38] and Habitat [3] simulator [42]. Following existing work [32, 40], the training process consists of two parts, *i.e.*, teacher forcing training on augmented data [44] and fine-tuning models using dagger training. We distribute training over 2 NVIDIA V100 GPUs for 3 days on average. A FORWARD action moves the agent forward by 0.25 meters and a TURN action turns by 15°. We use the same set of hyper-parameters as used in the VLN-CE [32] and show these values in Appendix. The U-Net used for extracting fine-grained features is pre-trained separately on RGB observations from Matterport3D scenes following existing work [19] and is frozen during training. The map size $m$ is set to 100. $\alpha$, $\beta$, and $\gamma$ in Equation (6) are set to 10 such that four reward terms are in the same order of magnitude at initialization. Progress threshold $\lambda_p$ is set to 0.8.

### 4.2 Comparisons with state-of-the-art methods

We compare our WS-MGMap with current state-of-the-art methods in Table 1 on both seen and unseen validation set of VLN-CE dataset. Our proposed WS-MGMap outperforms all methods that follow the same VLN-CE setting (*i.e.*no panoramic images) in terms of NE, OS, SR, and SPL, which demonstrates the effectiveness of the learned map representation. Besides, we also get comparable

---

[3]https://aihabitat.org/

Table 1: Comparison between our WS-MGMap and state-of-the-art methods on VLN-CE dataset. Methods marked with * use panoramic images. We highlight the best results among the methods that do not use panoramic images in all tables.

| | Val-Seen | | | | | Val-Unseen | | | | |
|---|---|---|---|---|---|---|---|---|---|---|
| | TL ↓ | NE ↓ | OS ↑ | SR ↑ | SPL ↑ | TL ↓ | NE ↓ | OS ↑ | SR ↑ | SPL ↑ |
| AG-CMTP* [8] | - | 6.60 | 56.2 | 35.9 | 30.5 | - | 7.90 | 39.2 | 23.1 | 19.1 |
| R2R-CMTP* [8] | - | 7.10 | 45.4 | 36.1 | 31.2 | - | 7.90 | 38.0 | 26.4 | 22.7 |
| HPN+DN* [31] | 8.54 | 5.48 | 53.0 | 46.0 | 43.0 | 7.62 | 6.31 | 40.0 | 36.0 | 34.0 |
| CWP-CMA* [25] | 11.47 | 5.20 | 61.0 | 51.0 | 45.0 | 10.90 | 6.20 | 52.0 | 41.0 | 36.0 |
| Seq2Seq [32] | 9.37 | 7.02 | 46.0 | 33.0 | 31.0 | 9.32 | 7.77 | 37.0 | 25.0 | 22.0 |
| CMA [32] | **9.26** | 7.12 | 46.0 | 37.0 | 35.0 | 8.64 | 7.37 | 40.0 | 32.0 | 30.0 |
| LAW [40] | 9.34 | 6.35 | 49.0 | 40.0 | 37.0 | 8.89 | 6.83 | 44.0 | 35.0 | 31.0 |
| SASRA [27] | 8.89 | 7.17 | - | 36.0 | 34.0 | **7.89** | 8.32 | - | 24.0 | 22.0 |
| CM2 [19] | 12.05 | 6.10 | 50.7 | 42.9 | 34.8 | 11.54 | 7.02 | 41.5 | 34.3 | 27.6 |
| WS-MGMap (Ours) | 10.12 | **5.65** | **51.7** | **46.9** | **43.4** | 10.00 | **6.28** | **47.6** | **38.9** | **34.3** |

Table 2: Results on VLN-CE challenge leaderboard. Methods marked with * use panoramic images.

| | Test-Unseen | | | | |
|---|---|---|---|---|---|
| Team | TL ↓ | NE ↓ | OS ↑ | SR ↑ | SPL ↑ |
| CWP-VLNBERT* [25] | 13.31 | 5.89 | 51 | 42 | 36 |
| CWP-CMA* [25] | 11.85 | 6.30 | 49 | 38 | 33 |
| WaypointTeam* [31] | 8.02 | 6.65 | 37 | 32 | 30 |
| VIRL_Team [32] | **8.85** | 7.91 | 36 | 28 | 25 |
| CM2 [19] | 13.85 | 7.74 | 39 | 31 | 24 |
| WS-MGMap (Ours) | 12.30 | **7.11** | **45** | **35** | **28** |

results compared with the methods that use panoramic images. This further verifies the importance of the WS-MGMap for helping agents perceive environments comprehensively even just using a camera sensor with a limited field of view. The detailed analyses are as follows.

We categorize existing methods into two types, *i.e.*, one for the methods that without building a map explicitly and one for that with a map representation. For the first type of methods (*e.g.*, Seq2Seq, CMA, LAW), our method outperforms them by a large margin. Specifically, compared with LAW, which is the strongest baseline among them, we increase the success rate from 40.0% to 46.9% and from 35.0% to 38.9% on val-seen and val-unseen, respectively. We attribute the improvement to the usage of our multi-granularity map, which represents environments explicitly for reducing the learning difficulty. As for the concurrent methods that are also with map representation (*e.g.*, SASRA and CM2), our WS-MGMap also brings a significant improvement against CM2, increasing success rate by 4.0% and 4.6% on val-seen and val-unseen, respectively. We suspect this is because the semantic map used by these methods provides insufficient information for VLN while our WS-MGMap solves this problem. All these results show the effectiveness of the proposed WS-MGMap, which learns a comprehensive map representation for VLN.

We note that our WS-MGMap even outperforms some baselines (*i.e.*, AG-CMTP, R2R-CMTP and HPN+DN) that use panoramic images on val-unseen, increasing success rate from 23.1%, 26.4% and 36.0% to 38.9%, respectively. This show the potential for replacing expensive panoramic camera with a common camera for VLN. We also note that our method has a relatively longer trajectory length. We argue that this is not a good metric for evaluating VLN performance because an unsatisfactory agent who always stops before it reaches the goal will also get a short trajectory length. A better alternative metric is SPL, which considers both episode length and success rate. In terms of SPL, our method significantly outperforms the existing methods who use the same settings with us (37.0% *v.s* 43.4% on val-seen and 31.0% *v.s* 34.3% on val-unseen).

**VLN-CE leaderboard.** We compare our WS-MGMap with prior work on the held-out test-unseen set used for VLN-CE leaderboard[4]. In Table 2, our method is leading among those that use the standard observation (no panoramas) following VLN-CE settings [32]. For a fair comparison, we only report the results whose manuscripts are publicly available to ensure that no strong prior knowledge (*e.g.*, exploring the environment in prior) is used.

---

[4]https://eval.ai/challenge/719/leaderboard/1966

Table 3: Ablation study on different granularity information for map representation.

| Map Type | Val-Unseen | | | | |
|---|---|---|---|---|---|
| | TL ↓ | NE ↓ | OS ↑ | SR ↑ | SPL ↑ |
| No Map | 11.66 | 7.20 | 36.8 | 26.5 | 21.4 |
| Fine-Grained Map | 10.11 | 7.11 | 41.1 | 31.6 | 28.2 |
| Semantic Map | 10.89 | 6.80 | 42.2 | 33.3 | 28.2 |
| Multi-Granularity Map (Ours) | **10.00** | **6.28** | **47.6** | **38.9** | **34.3** |

Table 4: Ablation study on weakly-supervised auxiliary task and localization ground-truth types.

| Auxiliary Task | GT Type | Val-Unseen | | | | |
|---|---|---|---|---|---|---|
| | | TL ↓ | NE ↓ | OS ↑ | SR ↑ | SPL ↑ |
| ✗ | - | 10.24 | 6.68 | 41.7 | 33.1 | 29.0 |
| ✓ | Hard | **9.36** | 6.36 | 43.6 | 35.2 | 32.3 |
| ✓ | Soft | 10.00 | **6.28** | **47.6** | **38.9** | **34.3** |

## 4.3 Ablation studies

**Effectiveness of multi-granularity map.** Our WS-MGMap contains both fine-grained details and semantic information about environments. We are interested to evaluate whether this different granularity information helps to represent environments for VLN. To this end, we construct two variants, *i.e.*, one only uses the fine-grained map to represent environments, and another one only uses semantic maps (combination of a semantic map projected from image segmentation results and our predicted global semantic map). All other settings are kept the same, including training losses and the auxiliary task. In Table 3, our WS-MGMap significantly outperforms these two variants, increasing success rate from 31.6% and 33.3% to 38.9%, respectively. These results show that both granularity information is important for VLN. Without the fine-grained map, agents are hard to recognize object attributes. Without the semantic map, agents are required to recognize all objects, including some common objects that can be represented in the semantic map, from fine-grained object details, which increases learning difficulty. In comparison, our WS-MGMap leverages complementary multi-granularity information for representing diverse objects. We also try a variant that directly takes raw RGB-D as input to perform VLN in an end-to-end manner. However, this variant drops success rate from 38.9% to 26.5%, indicating the importance of building a map explicitly for VLN.

**Effectiveness of weakly-supervised auxiliary task.** For teaching agents to learn a representative map that represents diverse objects accurately, we propose a weakly-supervised instruction-relevant object localization auxiliary task. To evaluate the effectiveness of this task, we remove the object localization loss in Equation (2), so that there is no auxiliary supervision signal for learning correspondence between maps and instructions. In Table 4, this variant performs worse than our WS-MGMap, success rate decreasing significantly from 38.9% to 33.1%. This shows the importance of the proposed weakly-supervised auxiliary task, which helps to learn map representation from the natural correspondence between map and instruction objects. The coarse ground truth used for this task is soft, *i.e.*, each map region is with a probability from 0 to 1. In this way, agents are required to figure out the degree of correlation between instructions and all map regions. An alternative is a hard binary ground-truth, *i.e.*, map regions whose region-path distance smaller than a threshold are considered positive and other regions are considered negative. In Table 4, this alternative performs better than the baseline that does not use our proposed auxiliary task but is worse than our soft localization ground-truth.

**Effect of different layer features for fine-grained map.** In this paper, we use a U-Net pre-trained for segmentation task as an RGB encoder to extract image fine-grained details. Existing network interpretability work [52, 28] points out that latent features from lower layers are dominated by color and texture concepts while features from higher layers present more object parts. To evaluate which types of features are more representative for describing environment fine-grained details, we project different layer latent features (*i.e.*, first layer, middle layer, last layer, and classification layer) to build fine-grained maps. In Table 5, using last layer features outperforms other variants in terms of almost all metrics. This is reasonable because the last layer features contain information from both the lower layer and higher layer because of the skip connection mechanism in U-Net, which make it more representative for representing environments. The features produced from the classification layer are

Table 5: Ablation study on projecting latent features from different layers to build fine-grained map.

| Projected Feature | Val-Unseen | | | | |
|---|---|---|---|---|---|
| | TL ↓ | NE ↓ | OS ↑ | SR ↑ | SPL ↑ |
| First Layer | 9.14 | 6.29 | 39.6 | 32.1 | 28.6 |
| Classification Layer | **9.05** | 6.49 | 40.2 | 33.3 | 29.6 |
| Last Layer | 10.00 | **6.28** | **47.6** | **38.9** | **34.3** |

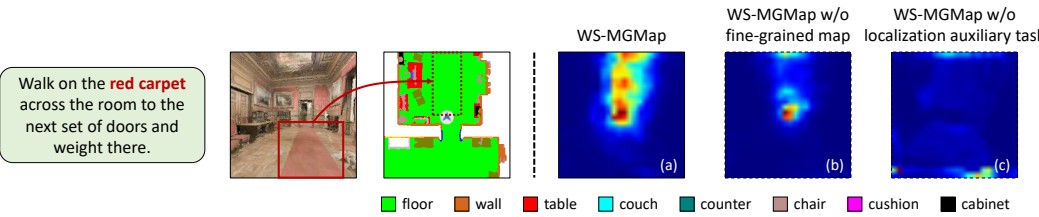

Figure 3: Visualization of instruction-relevant object localization results.

logits for semantic classification. These features are dominated by semantics information, providing less fine-grained environment details. We also find that an agent using middle layer features cannot be trained to convergence. We suspect it is because the spatial resolution of the middle features ($9 \times 9$) is too low such that the projected fine-grained map is sparse.

### 4.4 Visualization results

To evaluate whether the proposed multi-granularity map helps to represent diverse objects, we visualize the localization results $\hat{\mathbf{P}}$ from Equation (3) in Figure 3. Compared with the variant that does not contain a fine-grained map (b) and the variant without the weakly-supervised auxiliary task (c), our WS-MGMap (a) localizes instruction objects more precisely. Specifically, *red carpet* is incorrectly recognized as *floor* by a segmentation model. With fine-grained map, our WS-MGMap localizes it robustly. The variant without fine-grained map is confused to localize it because the semantic map can not represent *red* attribute and *carpet* object. The variant without the localization auxiliary task also performs worse because it is hard to establish the correspondence between map representation and diverse instruction-relevant objects. More results are shown in Appendix.

## 5 Discussion

**Limitations and future work.** Although our learned multi-granularity map helps for representing environments comprehensively and provides useful information to improve VLN performance, the semantic loss used in Equation (1) needs ground-truth semantic annotation, and our 2D top-down map is hard to handle the situation when agents go to another floor. Future work may explore to adapt 3D mapping technique [5] to our multi-granularity map. Besides, to improve the language-to-object grounding, exploiting commonsense to find relevant areas of instruction-mention objects could be an interesting future research direction. In addition, although our method shows promising performance in photo-realistic simulated data, it has not been thoroughly evaluated in real world. Directing adapting it to real environments may cause accidents such as breaking somethings and hitting pedestrian.

**Conclusion.** In order to solve the problem that current maps provide insufficient information for VLN task, we propose to gather different granularity information (*i.e.*, fine-grained details and semantic information) for map representation. Moreover, to make the map better represent diverse objects described in instructions, we propose a weakly-supervised auxiliary task for learning to localize instruction-relevant objects on the map with no need of extra localization annotation. Experimental results show that the proposed method significantly outperforms state-of-the-art methods on VLN-CE benchmark dataset. Qualitative results also show that the proposed multi-granularity map helps to localize instruction objects that are incorrectly classified by current maps.

## Acknowledgments

This work was partially supported by National Key R&D Program of China (No. 2020AAA0106900), National Natural Science Foundation of China (No. 62072190), Program for Guangdong Introducing Innovative and Enterpreneurial Teams (No. 2017ZT07X183).

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
