# Appendix for
# "Weakly-Supervised Multi-Granularity Map Learning for Vision-and-Language Navigation"

In the appendix, we provide more implementation details and experimental results of our WS-MGMap. We organize the appendix as follows.

- In Sec. A, we provide more architecture details on semantic hallucination module, map encoder, and object localization modules.
- In Sec. B, we provide more experimental details, *i.e.*, training settings and evaluation metrics.
- In Sec. C, we provide more analysis on instruction-relevant object localization results.
- In Sec. D, we provide more ablation results on semantic hallucination module.
- In Sec. E, we provide more ablation results on dagger training paradigm.
- In Sec. F, we provide more experimental results on RxR-Habitat dataset.
- In Sec. G, we provide more analysis on the predicted waypoints for VLN.
- In Sec. H, we provide more visualization examples on instruction-object ambiguity cases.

## A  More architecture details

The architecture details on semantic hallucination module, map encoder, and objection localization module for WS-MGMap (Figure 2 in main paper) are shown in Figures A and B. We follow the PyTorch [6] conventions to describe each layer, and the tensor shapes are represented in (C, H, W) notations. The meanings of each layer are as follows:

- **ConvBR**: a combination of nn.Conv2d, nn.BatchNorm2d and nn.ReLU layers with the input channels, output channels, kernel size, stride and padding augments.
- **TransConvBR**: a combination of nn.ConvTranspose2d, nn.BatchNorm2d and nn.ReLU layers with the input channels, output channels, kernel size, stride and padding augments.
- **Conv**: a nn.Conv2d layer with the input channels, output channels, kernel size, stride and padding augments.
- **AvgPool**: a nn.AvgPool2d layer with the the kernel size, stride and padding arguments.

In our experiments, the fine-grained map, global semantic map, and multi-granularity map are of different sizes (as shown in Figure A) for saving GPU memory. It is flexible to change their sizes, keeping all map sizes the same, by changing the stride of convolutional layers. For a brief description, in the main paper, we describe that all maps are of the same size $m \times m$. In Figure B, RGB encoder is an off-the-shelf ResNet18 [3] pre-trained on ImageNet [1]. Depth encoder is an off-the-shelf ResNet50 [3] pre-trained on point-goal navigation [10]. $L$ is the number of words in an instruction.

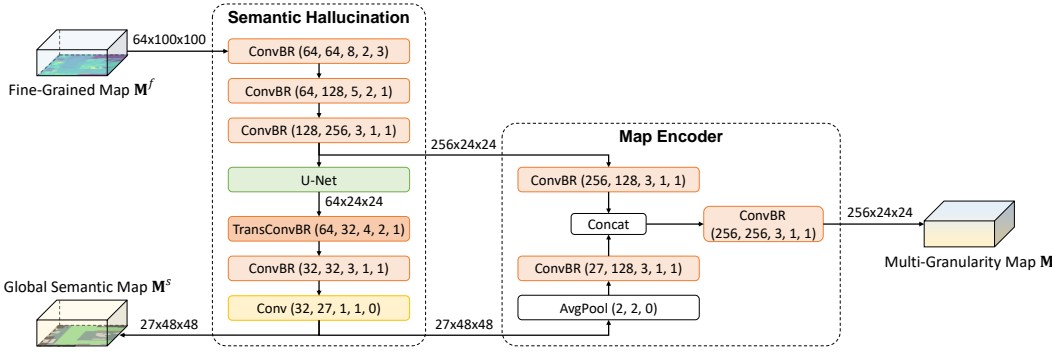

Figure A: Architecture of semantic hallucination module and map encoder.

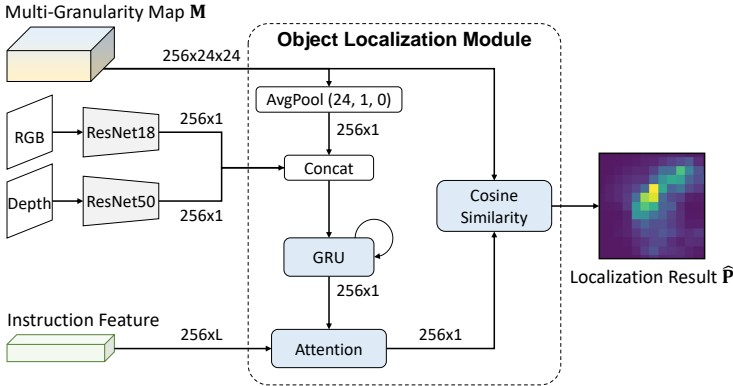

Figure B: Architecture of object localization module.

## B More experimental details

**Object categories predicted by hallucination module.** We select the same 27 common-seen object categories as CM2 [2] for the semantic hallucination module. We list all 27 object categories as follows: *{void, chair, door, table, cushion, sofa, bed, plant, sink, toilet, tv-monitor, shower, bathtub, counter, appliances, structure, other, free-space, picture, cabinet, chest-of-drawers, stool, towel, fireplace, gym-equipment, seating, clothes}*.

**More implementation details.** We use an Adam optimizer with a learning rate of 2.5e-4. For teacher-forcing training, we train on augmented trajectory data for 30 epochs. For dagger training, we collect 5,000 trajectories at each iteration (total 10 iterations). During the data collection process in $n^{th}$ iteration, the agent will take oracle action with probability $0.5^n$ and predicted action otherwise. At each iteration in dagger training, we train models on all collected trajectories for 4 epochs.

**More details on evaluation metrics.** We follow VLN-CE [5] to evaluate the navigation process in terms of success rate (SR), oracle success rate (OS), success weighted by path length (SPL), trajectory length (TL), and navigation error from goal (NE). A good agent should successfully navigate to the goal following the path described by an instruction. The details of each metric are described below.

- **SR**: ratio of agent calling STOP within a threshold distance (3 meters) of the goal in an allowed time step budget (500 steps).
- **OS**: ratio of agent reaching within a threshold distance (3 meters) of the goal.
- **SPL**: success rate weighted by path length, *i.e.*, $\text{SPL} = s \times d / max(d, \bar{d})$, where $s$ indicates the value of success rate, $d$ is the shortest geodesic distance from the starting point to the goal, $\bar{d}$ indicates the geodesic distance traveled by the agent.
- **TL**: average of agent trajectory length in meters.
- **NE**: average of geodesic distance from agent's final position to goal in meters.

## C More analysis on instruction-relevant object localization

We show additional visualization results of object localization (described in Sections 3.3 and 4.4 in the main paper) in Figure C. Our WS-MGMap method precisely localizes objects that are out of the segmentation category list (*e.g.*, refrigerator, shelf, railing in the first three rows, respectively) and objects that are specified by various attributes (*e.g.*, wooden slatted floor and white door in the last two rows, respectively). Note that the semantic map shown in the third column is not used by our methods. Instead, we build a multi-granularity map progressively, containing both fine-grained and semantic information about environments.

To quantitatively evaluate the instruction-relevant object localization performance, we measure IoU between ground-truth and predicted object locations. Specifically, we consider the 10% area with

the highest probability in 2D distribution $\mathbf{P}$ and $\hat{\mathbf{P}}$ (as described in Section 3.3) as ground-truth and predicted locations. The IoU values for our method, variant w/o fine-grained map, and variant w/o localization auxiliary task are 36.7%, 32.3%, and 6.9% respectively. These results further demonstrate that the proposed multi-granularity map and localization auxiliary task help agents localize instruction-relevant objects for the VLN task.

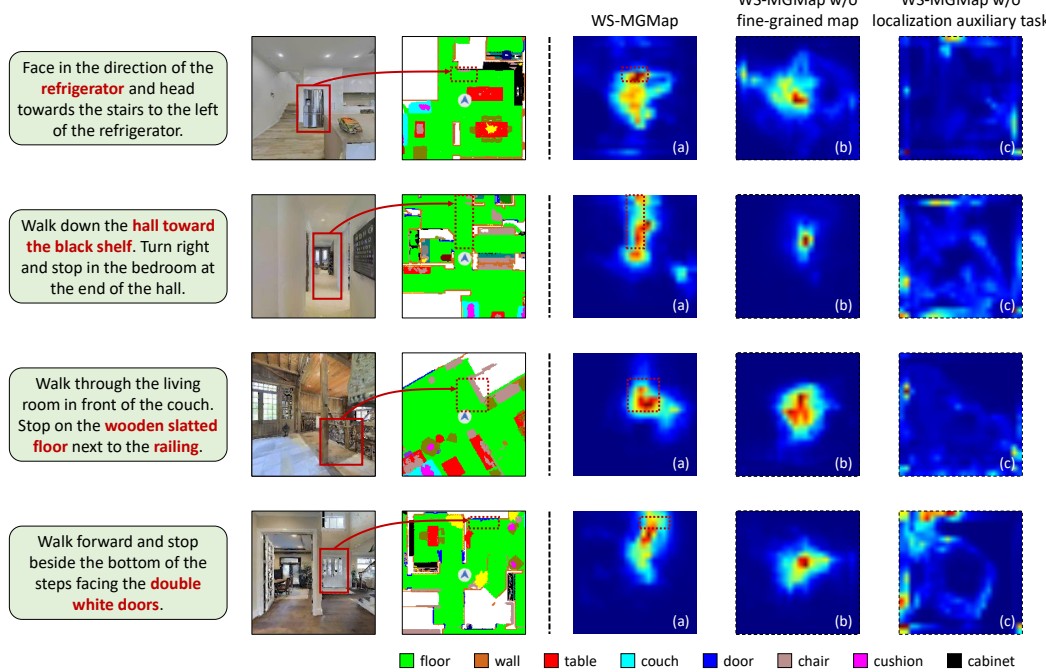

Figure C: Visualization of instruction-relevant object localization results.

## D More ablation study on semantic hallucination module

Following standard setting in VLN-CE task [5], we equip the agent with a camera with a limited field of view (the horizontal field-of-view is set to 90 degrees). In this sense, the agent can only capture RGB-D information about a small area in the environment. Motivated by existing works [2, 7], we design a hallucination module that helps the agent hallucinate the areas that are out of view range. To evaluate its effectiveness, we implement a variant that only predicts the semantic map within the field of view (i.e., w/o hallucination). From Table 1, this variant performs worse than our agent. These results show the importance of semantic hallucination for VLN.

Table 1: Ablation study on semantic hallucination module.

| Method | Val-Unseen | | | | |
| | TL ↓ | NE ↓ | OS ↑ | SR ↑ | SPL ↑ |
|---|---|---|---|---|---|
| w/o hallucination | 10.05 | 6.51 | 44.9 | 37.3 | 33.2 |
| w/ hallucination (Ours) | **10.00** | **6.28** | **47.6** | **38.9** | **34.3** |

## E More ablation results on DAgger training

During the second training stage, we follow existing works [5, 8, 4] to use Dagger [9] training techniques. As shown in these works, Dagger training helps to eliminate the negative effect of disconnection between training and testing caused by imitation learning. To evaluate its effectiveness, we conduct an experiment by replacing Dagger training with imitation learning. In Table 2, removing schedule sampling (i.e., using ground-truth trajectories at all times) drops the performance significantly.

Table 2: Ablation study on training paradigm.

| Method | Val-Unseen | | | | |
|---|---|---|---|---|---|
| | TL ↓ | NE ↓ | OS ↑ | SR ↑ | SPL ↑ |
| w/o DAgger Training | 7.9 | 7.61 | 30.0 | 24.6 | 22.5 |
| w/ DAgger Training (Ours) | **10.00** | 6.28 | **47.6** | **38.9** | **34.3** |

# F    More experimental results on RxR-Habitat dataset.

We leverage the model trained on R2R-Habitat and transfer it to test on RxR-Habitat English data split. We compare our methods with LAW [8], which is a competitive baseline in the VLN task. From Table 3, our method outperforms different variants of LAW on both val-seen and val-unseen data splits. It is worth noting that our model is directly transferred from R2R-Habitat to RxR-Habitat without finetuning. These results demonstrate the effectiveness and robustness of our method.

Table 3: Comparison on RxR-Habitat dataset (English language split).

| | Val-Seen | | | | | Val-Unseen | | | | |
|---|---|---|---|---|---|---|---|---|---|---|
| | TL ↓ | NE ↓ | OS ↑ | SR ↑ | SPL ↑ | TL ↓ | NE ↓ | OS ↑ | SR ↑ | SPL ↑ |
| LAW pano [8] | **6.27** | 12.07 | 17.0 | 9.0 | 9.0 | 4.62 | 11.04 | 16.0 | 10.0 | 9.0 |
| LAW step [8] | 7.92 | 11.94 | 20.0 | 7.0 | 6.0 | **4.01** | 10.87 | 21.0 | 8.0 | 8.0 |
| WS-MGMap (Ours) | 10.37 | **10.19** | **27.7** | **14.0** | **12.3** | 10.80 | **9.83** | **29.8** | **15.0** | **12.1** |

# G    More analysis on navigation episodes.

We show a navigation example in Figure D. An agent first perceives the environment by progressively building a multi-granularity map (semantic maps are shown in Figure D for a demonstration purpose only). Based on the multi-granularity map and instruction, the agent predicts a waypoint at every three time steps, which leads the agent to navigate following the instruction.

To quantitatively evaluate the quality of predicted waypoints, we report the percentage of predicted waypoints that locate within ground-truth path area, which is defined as 10% area with the highest probability in 2D distribution **P**. Compared with the variant w/o fine-grained map (57.0%) and the variant w/o localization auxiliary task (55.1%), the predicted waypoints from our method are more accurate, with 61.2% waypoints located within the ground-truth path area.

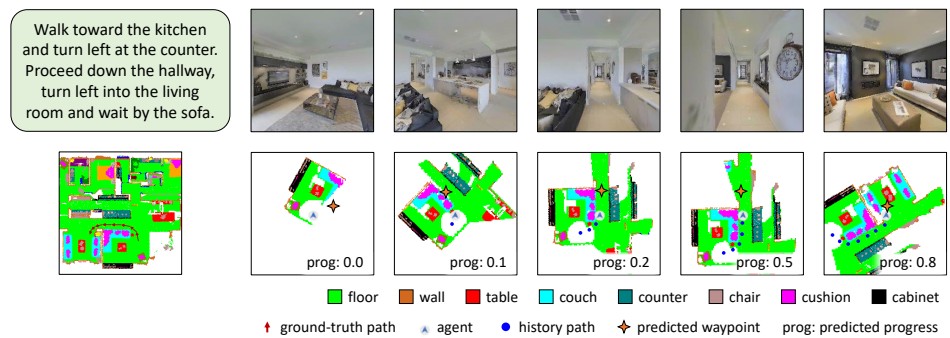

Figure D: A navigation example using our WS-MGMap on val-unseen data split.

# H    More visualization on instruction-object ambiguity.

As described in Introduction section, there exist instruction-object ambiguity cases (*e.g.*, the instruction object being a long bench but there are multiple different kinds of benches nearby) in VLN task. To quantitatively evaluate how often this type of instruction-object ambiguity occurs, we manually annotate these cases using a crowd-sourcing platform AMT. Quantitatively, there are

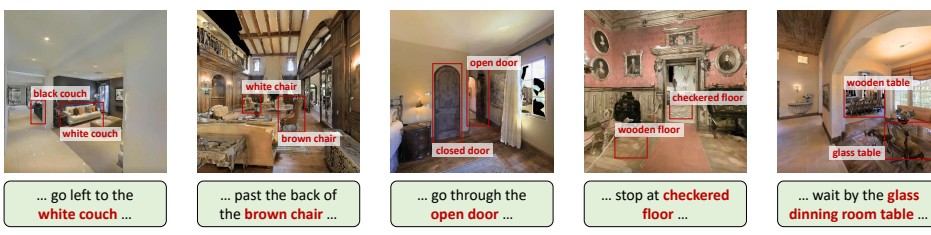

Figure E: Examples of instruction-object ambiguity cases.

51% instructions containing objects described by specific attributed words (e.g., wooden table). 33% trajectories of these instructions occur instruction-object ambiguity. We show some examples of such instruction-object ambiguity cases in Figure E. The first row shows the observation captured during navigation and the second row shows the object description in instructions. These ambiguity cases further demonstrate the necessity to build a multi-granularity map to include both object fine-grained details and semantic information for VLN task.