# OpenReview forum: "Weakly-Supervised Multi-Granularity Map Learning for Vision-and-Language Navigation"
_NeurIPS.cc/2022/Conference — NeurIPS 2022 Accept_

### Official Review · Reviewer_RkNa · 2022-07-10

**Rating:** 7
**Confidence:** 3
**Soundness:** 3 good
**Presentation:** 2 fair
**Contribution:** 3 good

**Summary:**

This paper presents an approach for vision-and-language navigation in continuous environments, centered around a semantic map which both aggregates and is used to predict object-centric information from the agent's observations. The approach uses fine-grained features extracted from a U-Net segmentation model applied to the agent's visual observations, as well as features for unseen parts of the environment predicted by a network supervised with ground-truth 3D info from the Matterport environment. This "multi-granularity" map is supervised using the ground-truth path of the agent through the environment, and used together with a recurrent, instruction-conditioned attention mechanism to predict waypoints for an off-the-shelf point-nav model to navigate to. The approach obtains the highest performance, by a substantial margin, among published VLN-CE leaderboard approaches that do not use panoramic image representations. Ablations show that each component of the approach helps pretty substantially.

**Questions:**

Q1) Could the authors clarify what sources of ground-truth semantic information from the Matterport3D simulator (beyond RGB and depth imagery) are used, and when they are used -- in training, inference, or both?

Q2) What parts of the model are updated during each of the two stages of training: teacher-forcing, and DAgger training? (e.g. is DD-PPO model held fixed during both of these, and the map prediction and object localization modules fine-tuned)?

Q3) What are the off-the-shelf FGB and depth encoders (line 197)?


**Limitations:**

I found the discussion of limitations in the paper somewhat lacking, although it should be easy to address given more space. In addition to discussing limitations due to object and environmental supervision (see above), it would also improve the paper to discuss more about possible directions that could improve the language-to-object grounding.

*Suggestions*
- Figure 1: the "bed" and "door" layers were unclear.
- The motivation in the intro confused me a bit about what type of object supervision is used.
- The object supervision in coarse localization was confusing to me. The localizer doesn't really seem to get any explicit object supervision, just closeness to the path -- and there doesn't seem to be flexibility for the model to choose how to focus the region attention using e.g. some sort of weakly-supervised or MIL loss. I think it might help to present this in a different way, e.g. as learning general info about the path (which may or may not be object-relevant).

*Minor suggestions*
- 112: "continued" -> "continuous"
- 149: "camera is with a" -> "the camera has a"
- 168: "reason over precise semantic information"
- 322: "carper" -> "carpet"


**Strengths And Weaknesses:**

*Strengths*

S1) The object representation and semantic map approach is well-motivated and well-executed, and the performance on the benchmark seems very strong.

S2) The paper performed a thorough set of experiments that helped me convince me all the decisions made in the approach (both levels of granularity in the semantic map; the localization task) were useful.

*Weaknesses*

W1) One weakness of the approach is that the full version depends on
ground-truth 3D semantic information from the Matterport3D simulator in training (although not in inference, I believe). If I understand Table 3 correctly, without this info (just using the fine-grained map with segmentation features) the approach performs underperforms most of the other non-panoramic approaches (CMA, LAW, CM2). Since this info is only used in training, I don't think it's a crucial weakness, but I feel the paper could be clearer about this limitation, in particular perhaps rephrasing or removing the motivating claim in the intro that contrasts its approach with "a naive solution that annotated a dataset ... with object classes and attributes for a segmentation model". (lines 40-41).

_Edit after response_: The response presented some compelling experiments showing that the proposed approach also improves when the semantic information is predicted by a pre-trained Faster R-CNN model, so I now feel like this is even less of a weakness.

W2) The approach, as I understand it, is quite similar to the approach of Blukis et al, "Mapping Navigation Instructions to Continuous Actions with Position-Visitation Prediction", CoRL 2018, which also uses object-centric semantic maps and waypoint prediction, to do language-conditioned navigation in a continuous environment. However, I don't think this is a crucial weakness as that approach was applied in a different environment with what seem to be much simpler visual observations (involving a discrete set of objects). It would strengthen this paper to contrast with that approach in the related work.

_Edit after response_: As the authors pointed out in their response, Blukis et al. does not use the multi-granular information that the proposed approach does.

W3) The writing could be clearer, see suggestions below.

---

> ### Author Response · Authors · 2022-08-02
> **Response to Reviewer RkNa**
>
> We thank you for your time and valuable comments. Below we answer the main concerns raised in the review and would be happy to provide further clarification if suitable.
>
> > **Q1) One weakness of the approach is that the full version depends on ground-truth 3D semantic information from the Matterport3D simulator in training.**
>
> Thanks for your comment. Our full version uses ground-truth 3D semantic information from the simulator for training but not for inference. We want to point out that some existing works (e.g., CM2 [a], MaAST[d]) also use this ground-truth information in training. During inference, we strictly follow the standard evaluation settings [b] for a fair comparison.
>
> We agree that not all datasets contain 3D semantic annotations. We will clarify this point in the limitation part.
>
> > **Q2) If I understand Table 3 correctly, without ground-truth semantic maps in training (just using the fine-grained map with segmentation features), the approach underperforms most of the other non-panoramic approaches (CMA, LAW, CM2).**
>
> We agree with the Reviewer that only using the fine-grained map performs slightly worse than some approaches, as shown in Tab. 3. We speculate this is due to the lack of high-level semantic information about the environment. However, we want to argue that we could easily obtain this semantic information even if the ground-truth semantic map annotations are not available for training.
>
> Specifically, we use a frozen pretrained segmentation model (e.g., Faster-RCNN) to segment the RGB images. Then we project the predicted segmentation results on a top-down map to build a semantic map. We use this map to replace the global semantic map in our method. In this sense, no semantic annotations are needed for training.
>
> Although the prediction from the pretrained segmentation model is imperfect, it provides complementary high-level information for our fine-grained map. We concatenate them to build a multi-granularity map. We conducted experiments on this variant, keeping all other settings the same except for the way of constructing a semantic map. The results on val-unseen data split are as below:
>
> | Method | TL ↓ | NE ↓ | OS ↑ | SR ↑ | SPL ↑ |
> |:---|:---:|:---:|:---:|:---:|:---:|
> | CMA [b] | 8.64 | 7.37 | 40.0 | 32.0 | 30.0 |
> | LAW [e] | 8.89 | 6.83 | 44.0 | 35.0 | 31.0 |
> | CM2 [a] | 11.54 | 7.02 | 41.5 | 34.3 | 27.6 |
> | Ours w/o semantic annotations | 11.0 | 6.45 | 44.6 | 36.7 | 32.1 |
> | Ours w/ semantic annotations | 10.00 | **6.28** | **47.6** | **38.9** | **34.3** |
>
> From the results, the variant without semantic annotations for training outperforms current non-panoramic approaches. This demonstrates the flexibility of our method to be adapted to the dataset without semantic annotations. For the datasets with semantic annotations (e.g., R2R, RxR), our method can exploit these annotations to further improve the performance.
>
> > **Q3) The approach is quite similar to the approach of Blukis et al, "Mapping Navigation Instructions to Continuous Actions with Position-Visitation Prediction", CoRL 2018. It would strengthen this paper to contrast with that approach in the related work.**
>
> Thanks for advising. Instead of only projecting a feature map generated from ResNet to construct the map as done in [c], we combine both fine-grained map and semantic map to generate a multi-granularity map. These two types of maps provide complementary object information to perceive a more complex photo-realistic environment. We will discuss that approach in the related work.
>
> > **Q4) Could the authors clarify what sources of ground-truth semantic information from the Matterport3D simulator are used, and when they are used?**
>
> We only use the semantic annotation from the simulator during training. In testing, we only need to obtain RGB and depth observation, which is a standard setting for evaluation [a,b]. Consequently, our evaluation is fair with the existing works.
>
> > **Q5) What parts of the model are updated during each of the two stages of training: teacher-forcing, and DAgger training?**
>
> The RGB and depth encoders, segmentation pre-trained model (a U-Net), and DD-PPO local policy are fixed during both two stages. All the other networks (namely semantic hallucination module, map decoder, Bi-LSTM, object localization module, progress predictor, and waypoint predictor) will be updated during both two stages. We will clarify these details in the revised paper.
>
> > **Q6) What are the off-the-shelf RGB and depth encoders?**
>
> Following existing paper VLN-CE [b], the off-the-shelf RGB and depth encoders are instantiated by a ResNet50 pretrained on ImageNet and a ResNet-50 trained to perform point-goal navigation, respectively. We will add these details in the revised paper.

---

> > ### Author Response · Authors · 2022-08-02
> > **Response to Reviewer RkNa (part2)**
> >
> >
> > > **Q7) In addition to discussing limitations due to object and environmental supervision, it would also improve the paper to discuss more possible directions that could improve the language-to-object grounding.**
> >
> > Thanks for your suggestions. One possible direction to improve the language-to-object grounding is exploiting commonsense (e.g. ConceptNet) to find relevant objects of the instruction-mention objects. In this way, the agent can localize instruction-mention objects by finding relevant objects. We will add more discussions about possible future directions in the revised paper.
> >
> > **Reference:**
> >
> > [a] Cross-modal Map Learning for Vision and Language Navigation. CVPR 2022.
> >
> > [b] Beyond the Nav-Graph: Vision-and-Language Navigation in Continuous Environments. ECCV 2020.
> >
> > [c] Mapping Navigation Instructions to Continuous Actions with Position-Visitation Prediction. CoRL 2018.
> >
> > [d] MaAST: Map Attention with Semantic Transformersfor Efficient Visual Navigation. ICRA 2021.
> >
> > [e] Language-Aligned Waypoint (LAW) Supervision for Vision-and-Language Navigation in Continuous Environments. EMNLP 2021.

---

### Official Review · Reviewer_eYGK · 2022-07-11

**Rating:** 7
**Confidence:** 4
**Soundness:** 3 good
**Presentation:** 3 good
**Contribution:** 3 good

**Summary:**

## Post Rebuttal

I thank the authors for their response. My main concern was addressed and the zero-shot evaluation on RxR is helpful. I encourage the authors to include a full evaluation on RxR. I have increased my rating.

## Pre-rebuttal
This paper presents a method for building an environment map towards the end of performing natural language instruction following in continuous environments (VLN-CE). In the proposed method, the map has multiple granularities of information. A coarse granularity with features projected from UNet, and a global semantic map created by hallucinating semantic labels for the entire environment based on the coarse granularity map.

The encourage the map contain features relevant to instruction following, the authors also propose an additional objective that predicts the location of the ground-truth path on the environment map. This predicted path location is then used to attend to the map.

The method is evaluated on the VLN-CE dataset and outperforms all existing single camera methods. It is also competitive with panoramic methods.

**Questions:**

See weaknesses

**Limitations:**

Yes

**Strengths And Weaknesses:**

### Strengths

The proposed method is well motivated. It makes sense to have multiple levels of information in a semantic map and to augment what is given to the agent based on the instruction.

The method works well. VLN-CE is a challenging task and a +4.5% SR increase is meaningful.

The ablation studies are well-done and provide useful insight.

### Weaknesses

No evaluation on RxR-Habitat (https://ai.google.com/research/rxr/habitat). This uses the more recent RxR dataset that fixed some of the issues with the original VLN dataset and thus evaluation on this dataset would be very helpful in evaluated the method.

The intro to the task should cite + mention VLN-CE since that's the variant of the task being studied (and the less common one). I recommend referring to the task as VLN-CE to avoid confusion.

L28-29: We don't know why humans (or any animal for that matter) build cognitive maps. Neither Tolman nor O'Keefe claim to know why cognitive maps were learned to be built (other than that they are useful for survival). They are an explanation of observed behavior, seen, but we don't know why that mechanism exists other than it's useful for survival (and thus is something that was learned implicitly). I recommend weakening this statement.

Self-supervised map representation learning via object location task is not a correct name. The objective is not self-supervised. Self-supervised implies that the learning objective is valid regardless of the dataset source (i.e. training data or data we'd see at evaluation). Ground-truth paths are certainly not part of the data we'd see at evaluation so this method doesn't fit. The task is possibly object localization with pseudo-labels (so weakly supervised), but those pseudo-labels are soft labels for predicting the ground-truth path. Since the motivation comes from object localization, that part of the name is probably fine, but the self-supervised part isn't.

Overall, I think this paper presents a meaningful contribution and am happy to increase my rating if these concerns are addressed.

---

> ### Author Response · Authors · 2022-08-02
> **Response to Reviewer eYGK**
>
> We thank you for your time and valuable comments. Below we answer the main concerns raised in the review and would be happy to provide further clarification if suitable.
>
> > **Q1) No evaluation on RxR-Habitat. Evaluation of this dataset would be very helpful in evaluating the method.**
>
> We agree that RxR-Habitat is very helpful for evaluating our method. In the rebuttal period, due to the time limit and large scale of this dataset, we are hard to retrain models in three languages of RxR-Habitat dataset. In this sense, we leverage the model trained on R2R-Habitat and transfer it to test on RxR-Habitat English data split. The results are shown in the table below. We compare our methods with LAW [a], which is a competitive baseline in the VLN task. We report the results below:
>
> * Table A: Results on RxR-Habitat val-seen data split.
>
> | Method | TL ↓ | NE ↓ | OS ↑ | SR ↑ | SPL ↑ |
> |:---|:---:|:---:|:---:|:---:|:---:|
> | LAW pano [a] | 6.27 | 12.07 | 17.0 | 9.0 | 9.0 |
> | LAW step [a] | 7.92 | 11.94 | 20.0 | 7.0 | 6.0 |
> | Ours | 10.37 | **10.19** | **27.7** | **14.0** | **12.3** |
>
> * Table B: Results on RxR-Habitat val-unseen data split.
>
> | Method | TL ↓ | NE ↓ | OS ↑ | SR ↑ | SPL ↑ |
> |:---|:---:|:---:|:---:|:---:|:---:|
> | LAW pano [a] | 4.62 | 11.04 | 16.0 | 10.0 | 9.0 |
> | LAW step [a] | 4.01 | 10.87 | 21.0 | 8.0 | 8.0 |
> | Ours | 10.80 | **9.83** | **29.8** | **15.0** | **12.1** |
>
> From the results, our method outperforms different variants of LAW on both val-seen and val-unseen data splits. It is worth noting that our model is directly transferred from R2R-Habitat to RxR-Habitat without finetuning. These results demonstrate the effectiveness and robustness of our method. We believe the performance could be further improved when the model is trained and tested on the RxR-Habitat. We will train and evaluate the complete dataset in the future.
>
> > **Q2) The intro to the task should cite + mention VLN-CE since that's the variant of the task being studied. I recommend referring to the task as VLN-CE to avoid confusion.**
>
> Thanks for your suggestions. We will update the manuscript to make it more clear.
>
> > **Q3) L28-29: We don't know why humans build cognitive maps. Neither Tolman nor O'Keefe claims to know why cognitive maps were learned to be built. I recommend weakening this statement.**
>
> Thanks for your reminding. We agree with the Reviewer that why and how humans build map-like representations are still unknown. We will weaken this statement in the revised paper.
>
> Although the mechanism of human mapping is unclear yet, in robotics, there exist many methods of building map (e.g., occupancy map [b], semantic map [c], topological map [d]) for navigation task. These methods achieve gratifying results. Inspired by these works, we consider to construct a similar map-like representation to perceive the environment. Meanwhile, to make the map better represent the instruction-relevant objects for VLN task, we propose a multi-granularity map and an instruction-relevant object localization auxiliary task to improve the VLN performance.
>
> We will carefully revise the paper to make our motivation more accurate and clear.
>
> > **Q4) Self-supervised map representation learning via object location task is not a correct name. The weakly-supervised part is probably fine.**
>
> Thanks for your valuable suggestions. We agree with the Reviewers that the term "weakly supervised" is more accurate. We will replace the "self-supervised" description with "weakly supervised" carefully in the revised paper to make it more clear and accurate.
>
> **Reference:**
>
> [a] Language-Aligned Waypoint (LAW) Supervision for Vision-and-Language Navigation in Continuous Environments. EMNLP 2021.
>
> [b] Learning to Explore using Active Neural SLAM. ICLR 2020.
>
> [c] Object Goal Navigation using Goal-oriented Semantic Exploration. NeurIPS 2020.
>
> [d] Topological Planning with Transformers for Vision-and-Language Navigation. CVPR 2021.

---

> > ### Author Response · Authors · 2022-08-06
> > **Sincerely Look Forward to Your Post-rebuttal Feedback!**
> >
> > Thanks again for your insightful suggestions and comments. As the deadline for discussion is approaching, we are glad to provide any additional clarifications that you may need.
> >
> > In our previous response, we have added evaluation results on RxR-Habitat dataset and carefully revised the paper to complement your suggestions. We summarize our responses with regard to the following aspects:
> >
> > * We have added evaluation results on RxR-Habitat dataset. Our method achieves state-of-the-art performance under non-panoramic settings, which further demonstrates the merit of our method (Q1).
> > * We have carefully revised the paper to make it more clear. The revisions include the following points. 1) We have mentioned and cited VLN-CE in Introduction (Q2).  2) We have weakened the statement of human behavior in the Introduction (Q3). 3) We have changed "self-supervised" to "weakly-supervised" throughout the paper, which is also recognized by Reviewer kB3d in the latest response (Q4). All the revisions are highlighted in red in the revised submission.
> >
> > We hope that the provided new experiments and revisions could convince you of the merits of our work.  Please do not hesitate to contact us if there are other clarifications or experiments we can offer.
> >
> > Thank you for your time again!
> >
> > Best, Authors

---

### Official Review · Reviewer_kB3d · 2022-07-11

**Rating:** 7
**Confidence:** 4
**Soundness:** 3 good
**Presentation:** 3 good
**Contribution:** 3 good

**Summary:**

This paper tackles the Visual & Language Navigation task. The paper proposes a new "multi-granular" map representation for encoding coarse-grained (semantic) and fine-grained (textures, colors, etc) details about the 3D space. It also proposes a self-supervised loss to localize "instruction-relevant" objects on the map. Experiments are performed on the VLN-CE benchmark on MP3D dataset. The work demonstrate strong improvement over prior methods for VLN-CE while using limited field-of-view sensing (as opposed to panoramic sensing).

**Questions:**

# Novelty of multi-granular maps

- Is the proposed mapping method achieving something different compared to prior work [1, 2, 3, 4]?
- If so, can the authors quantitatively compare against map representations from [2] (and maybe even [1, 3, 4]) and verify it is indeed better? That is, replace the proposed multi-granular maps while keeping the auxiliary task and waypoint navigation as it is.

# Novelty of auxiliary task

The method from [5] picks waypoints along the demonstration trajectory, and places gaussians at each waypoint to obtain the target heatmap for prediction.
- Is the heatmap P in L177-182 different from the heatmap in [5]?
- If they are indeed different (i.e., P focuses more on instruction-relevant objects rather than waypoints), could the authors replace the heatmap P with the heatmap from [5] and verify that the proposed method is indeed better? (note: the multi-granular maps and waypoint navigation pipeline remain the same)
- Is the auxiliary task really self-supervised? That is, is it learning to localize instruction-relevant objects? Is the heatmap from [5] not resulting in localizing instruction-relevant objects in that case?

# Other clarifications in approach and experiments
- Could the authors clarify the questions raised in the weaknesses?

[1] **MapNet:** Henriques, Joao F., and Andrea Vedaldi. "Mapnet: An allocentric spatial memory for mapping environments." proceedings of the IEEE Conference on Computer Vision and Pattern Recognition. 2018.
[2] **Semantic MapNet:** Cartillier, Vincent, et al. "Semantic mapnet: Building allocentric semantic maps and representations from egocentric views." Proceedings of the AAAI Conference on Artificial Intelligence. Vol. 35. No. 2. 2021.
[3] **Cognitive Mapping and Planning**: Gupta, Saurabh, et al. "Cognitive mapping and planning for visual navigation." Proceedings of the IEEE conference on computer vision and pattern recognition. 2017.
[4] **Geometry-Aware RNNs**: Cheng, Ricson, Ziyan Wang, and Katerina Fragkiadaki. "Geometry-aware recurrent neural networks for active visual recognition." Advances in Neural Information Processing Systems 31 (2018).
[5] **Cross-Modal Map Learning**: Georgakis, Georgios, et al. "Cross-modal Map Learning for Vision and Language Navigation." Proceedings of the IEEE/CVF Conference on Computer Vision and Pattern Recognition. 2022.

**Limitations:**

Yes. They have been sufficiently addressed.

**Strengths And Weaknesses:**

# ----------------------- Post-rebuttal update --------------------
The author's responses during the rebuttal were very convincing and addressed my concerns. The main weaknesses pre-rebuttal were the novelty of the multi-grain map representation and the auxiliary loss. In response, the authors compared and contrasted with the cited works, and even showed quantitative benefits in relation to them. I've raised my rating to 7 (accept).

# ----------------------- Pre-rebuttal review --------------------
# Strengths

- The paper is well motivated in building multi-granular maps for wholistic object representations and grounding instructions to relevant objects on the map.
- The paper writing is good and it was easy to read and understand the ideas and contributions.
- Experiments are well designed. A lot of relevant and appropriate baselines have been compared with. The experimental results show good improvements over comparable methods. Ablation studies confirm the value of the contributions. Following traditions from prior work on this task, the authors also present results on an online leaderboard, which verifies the claims made under standardized settings.

# Weaknesses
## Novelty of multi-granular maps for embodied AI
- The proposed multi-granular maps does the following: (1) use image segmentation features to capture fine-grained details, (2) project the features to get a top-down 2.5D fine-grained map, (3) obtain coarse-grained segmentation maps with potential expansion beyond observed cells, and (4) jointly merging fine-grained and coarse-grained to obtain final representations. This particular pipeline itself appears novel (particularly in VLN).
- However, prior work [1, 2, 3, 4] has extensively studied mapping in high-dimensional feature spaces (as opposed to occupancy / semantic maps). These have not been discussed in Sec. 2.2.
- More importantly, they might be doing exactly what the proposed method intends to do. The general idea is to get visual feature maps of the egocentric inputs, and project them in a geometric / learned fashion to top-down maps / voxel grids. That is, the multi-grained map may be obtained implicitly here as opposed to explicitly as done in the Fig. 2.
- For example, Semantic MapNet [2] builds such maps from visual inputs in the encoder, and decodes them to obtain semantic maps. In L145-146, the authors suggest that the last layers of an image segmentation model (Unet) contain a good mix of textural and semantic features. It is reasonable to expect the same to hold true for the last layers of Semantic MapNet to encode something similar, but in the top-down map space.


## Novelty of auxiliary task
- L174-176: "The principle for generating coarse ground-truth is that the regions closer to the path have a higher probability of containing instruction-relevant objects." --- It is unclear to me whether this supervision is resulting in localizing "instruction-relevant" objects.
- This objective could also plausibly be viewed as a form of imitation learning for waypoint prediction (like [5]). For example, in Figure 3, the model could be learning to predict the locations of future waypoints to follow instead of referring to the carpet (i.e., the carpet being along the waypoints could just be a coincidence).
- If the model were to just predict future waypoints, then the method will no longer be self-supervised. The self-supervised component only refers to how "instruction-relevant" objects are localized without explicit supervision.


## Need clarifications in approach and experiments
- L152-156 - is hallucination of maps beyond the field of view helpful quantitatively?
- Eqn 1 - is it a pixelwise loss?
- L173 "Because we do not know the exact locations of these instruction objects" --- how often does this type of instruction-object ambiguity occur? For example, the instruction object being a long-bench and there are multiple benches nearby. I can imagine that in quite a few cases, the objects might be unique and can be localized accurately.
- L177-184 - does P remain the same for all time-steps of the trajectory? If so, is it possible for the model to just ignore visual inputs and map the fixed language instruction to the fixed P (regardless of the agent states)?
- In Tab. 3 ablation study - do all rows benefit from the auxiliary task? For example, the result without auxiliary task in Tab. 4 is close to semantic map (row 3 in Tab. 3). If semantic map does not use auxiliary task, then the gain from multi-granular representations itself is limited.
- **[Minor point, not weakness]** L231 - how is the DAgger training performed without a human in the loop? Might be useful to add a section in the supplementary to explain this.
- **[Minor point, not weakness]** In Tab. 5 ablation study - classification layer logits are better than semantic map from Tab. 3. Could the authors comment on this?


# Typos and grammatical errors
L11 - "Moreover, we propose" -> "We further propose"
L64 - followings -> follows
L87 - "disables the agent to ground … well"  -> "prevents the agent from effectively grounding …"
L114 - "access to predefined" -> "access the predefined"
L122 - "larger number" -> "large amount"
L164 - procuding -> producing
L168 - "reason precise semantics" -> "reason precisely about semantic"
L231 - "teaching force" -> "teacher forcing"
L322 - "carper" -> "carpet"



[1] **MapNet:** Henriques, Joao F., and Andrea Vedaldi. "Mapnet: An allocentric spatial memory for mapping environments." proceedings of the IEEE Conference on Computer Vision and Pattern Recognition. 2018.
[2] **Semantic MapNet:** Cartillier, Vincent, et al. "Semantic mapnet: Building allocentric semantic maps and representations from egocentric views." Proceedings of the AAAI Conference on Artificial Intelligence. Vol. 35. No. 2. 2021.
[3] **Cognitive Mapping and Planning**: Gupta, Saurabh, et al. "Cognitive mapping and planning for visual navigation." Proceedings of the IEEE conference on computer vision and pattern recognition. 2017.
[4] **Geometry-Aware RNNs**: Cheng, Ricson, Ziyan Wang, and Katerina Fragkiadaki. "Geometry-aware recurrent neural networks for active visual recognition." Advances in Neural Information Processing Systems 31 (2018).
[5] **Cross-Modal Map Learning**: Georgakis, Georgios, et al. "Cross-modal Map Learning for Vision and Language Navigation." Proceedings of the IEEE/CVF Conference on Computer Vision and Pattern Recognition. 2022.

---

> ### Author Response · Authors · 2022-08-02
> **Response to Reviewer kB3d**
>
> We thank you for your time and valuable comments. Below we answer the main concerns raised in the review and would be happy to provide further clarification if suitable.
>
> > **Q1) Comparisons with prior works [a,b,c,d].**
>
> We have cited and discussed the four suggested papers in the revised version. We use mapping in high-dimensional feature spaces (named fine-grained map) for a fundamentally different goal compared with these works. Directly exploiting the maps in [b] for VLN performs significantly worse than our method. We in fact have already provided these comparison results in Tables 3 and 4. The detailed comparisons are shown as follows.
>
> **1. The goals of using fine-grained map are fundamentally different**. In VLN task, there exist diverse objects with different attributes (typically hundreds of object categories) in the instruction and environment. Being aware of these object locations is critical for agents to find the navigation path. We aim to use the fine-grained map for providing detailed object information (e.g., color and material) to represent diverse instruction-relevant objects. In contrast, existing works use this map for localizing robot [a], predicting semantic map with limited object category [b], navigating to a few target objects [c,e], and 3D reconstructing objects [d]. **None of these works have investigated how to build a map for representing diverse objects with different attributes.**
>
> **2. To achieve our goal of representing diverse objects in a map, we propose the following two innovations upon existing works.**
>
> **2.1 Constructing a multi-granularity map.** Instead of only using a fine-grained map as done in [a,b,c,d] or only using a semantic map, we are the first to combine them to generate a multi-granularity map for the VLN task. These two types of maps provide complementary object information, i.e., a fine-grained map provides detailed object information and attributes while a semantic map provides commonly seen object category information. From the table below, only exploiting a fine-grained map built in [b] or semantic map significantly drops the performance. These results have been already provided in Tab. 3 in the paper.
>
> | Map Type | TL ↓ | NE ↓ | OS ↑ | SR ↑ | SPL ↑ |
> |:---|:---:|:---:|:---:|:---:|:---:|
> | Fine-grained Map [b] | 10.11 | 7.11 | 41.1 | 31.6 | 28.2 |
> | Semantic Map | 10.89 | 6.80 | 42.2 | 33.3 | 28.2 |
> | Fine-grained Map + Semantic Map (Ours) | **10.00** | **6.28** | **47.6** | **38.9** | **34.3** |
>
>
>
> **2.2 Constructing an auxiliary task for localizing diverse objects.**
> To provide supervision for learning such a multi-granularity map to represent diverse objects, we propose an instruction-relevant object localization auxiliary task. As a comparison, [b] uses a semantic map with limited categories as supervision, while [c] uses final navigation actions as supervision for learning map representation. These supervisions can not explicitly teach models to represent diver objects. We have also tried a variant that only uses these two types of supervision (i.e., using semantic loss in Eq. (1) and navigation loss in Eq. (5)). From the table below, this variant performs worse, while our proposed auxiliary task improves the performance significantly. These results have been already provided in Tab. 4 in the paper.
>
> | Supervision Signal | TL ↓ | NE ↓ | OS ↑ | SR ↑ | SPL ↑ |
> |:---|:---:|:---:|:---:|:---:|:---:|
> | Semantic Loss [b] + Navigation Loss [c] | 10.24 | 6.68 | 41.7 | 33.1 | 29.0 |
> | Ours | **10.00** | **6.28** | **47.6** | **38.9** | **34.3** |
>
>
>
>
> > **Q2) The multi-grained map may be obtained implicitly in prior works [a,b,c,d].**
>
> Directly using these implicit multi-grained maps achieves inferior performance. This is because they cannot represent diverse objects in VLN due to **the lack of high-level semantic information and appropriate supervision**. Upon these maps, we further 1) propose to explicitly build a multi-grained map and 2) propose an auxiliary task for learning diverse object representation. Experimental results show these two techniques bring significant improvement (please refer to the two tables in Q1).
>
> We will include and discuss these works in Sec. 2.2 in the revised paper.

---

> > ### Author Response · Authors · 2022-08-02
> > **Response to Reviewer kB3d (part2)**
> >
> >
> > > **Q3) Comparisons with waypoint prediction (like [e]).**
> >
> > Our auxiliary task is fundamentally different from waypoint prediction in the aspect of supervision signal.
> >
> > Specifically, the waypoint prediction task only requires predicting one or several discrete points on the ground-truth path. Taking CM2 [e] as an example, it aims to predict 10 waypoints, each of which is represented as a Gaussian distribution. Combining these 10 Gaussian distributions obtains a **waypoint heatmap** $P_w$ as the prediction target. However, this heatmap cannot represent the potential location of instruction-relevant objects. Because these objects are very likely to appear in the area between two adjacent waypoints, but the waypoint heatmap [e] has a low value in these areas. In contrast, we build an **object heatmap** $P_o$, which follows the principle that the regions closer to the path have a higher probability of containing these objects. Our heatmap provides more precise supervision for object localization.
> >
> > We have conducted an experiment that replaces our proposed object heatmap $P_o$ with waypoint heatmap $P_w$ [e]. From the table below, the waypoint heatmap performs worse than ours. Its instruction-relevant object localization accuracy is also worse than ours (see **Q4** for experimental details and localization IoU). These results demonstrate the effectiveness of our proposed auxiliary task.
> >
> > | Heatmap Type | TL ↓  | NE ↓    | OS ↑ | SR ↑  | SPL ↑ |
> > |:--|:--:|:--:|:--:|:--:|:--:|
> > | Waypoint Heatmap [e] | 11.80 | 6.53 | 45.4 | 34.4 | 29.3 |
> > | Object Heatmap (Ours) | **10.00** | **6.28** | **47.6** | **38.9** | **34.3** |
> >
> > > **Q4) Is the supervision of the auxiliary task learn to localize instruction-relevant objects? What about the heatmap from [e]?**
> >
> > To evaluate the localization performance, we manually annotate the accurate position of instruction-relevant objects in 1/5 trajectories (total 360 trajectories) in the val-unseen split. Specifically, we localize the instruction-relevant objects in the RGB images, and then project them to the top-down map. We consider the 20% area with the highest score on the prediction map as the predicted location. From the table below, our method achieves 0.33 IoU, outperforming the variant that uses waypoint heatmap [e] as supervision and the variant without an auxiliary task.
> >
> > | Auxiliary Task | Heatmap Type | IoU ↑ |
> > |:---:|:---:|:---:|
> > | X | None | 0.0654 |
> > | Y | Waypoint Heatmap [e] | 0.2241 |
> > | Y | Object Heatmap (Ours) | **0.3314** |
> >
> > > **Q5) Is hallucination of maps beyond the field of view helpful quantitatively?**
> >
> > The hallucination helps agents explore the environment more effectively, which is proved by existing works [e,f]. To evaluate its effectiveness, we implement a variant that only predicts the semantic map within the field of view (i.e., w/o hallucination). From the table below, this variant underperforms our agent.
> >
> > | Method | TL ↓ | NE ↓ | OS ↑ | SR ↑ | SPL ↑ |
> > |:---|:---:|:---:|:---:|:---:|:---:|
> > | w/o hallucination | 10.05 | 6.51 | 44.9 | 37.3 | 33.2 |
> > | w hallucination (Ours) | **10.00** | **6.28** | **47.6** | **38.9** | **34.3** |
> >
> > > **Q6) Eqn 1 - is it a pixelwise loss?**
> >
> > Yes, exactly. We will update the paper to make it more clear.
> >
> > > **Q7) How often does instruction-object ambiguity occur?**
> >
> > These ambiguous cases actually occur quite often. Quantitively, 1) there are **51%** instructions containing objects described by specific attributed words (e.g., wooden table). 2) **33%** trajectories of these instructions occur instruction-object ambiguity. We obtain these numbers by manual annotations in the AMT crowdsourcing platform.

---

> > > ### Author Response · Authors · 2022-08-02
> > > **Response to Reviewer kB3d (part3)**
> > >
> > > > **Q8) Does P remain the same for all time-steps of the trajectory?**
> > >
> > > No, P is rotated at each time step to guarantee that the agent is facing upward. Consequently, it is impossible for the model to predict P just from instruction because the instruction does not contain agent heading information.
> > >
> > > > **Q9) In Tab. 3 ablation study - do all rows benefit from the auxiliary task? For example, the result without the auxiliary task in Tab. 4 is close to the semantic map (row 3 in Tab. 3). If the semantic map does not use the auxiliary task, then the gain from multi-granular representations itself is limited.**
> > >
> > > All rows in Tab. 3 have used the auxiliary task (except for the "No Map" row because it is hard to localize objects without a map). With the same experimental settings, just replacing the multi-granularity map with the semantic map (row 3 in Tab. 3) or fine-grained map (row 2 in Tab. 3) significantly drow the performance. This demonstrates the importance of the proposed multi-granularity map.
> > >
> > > > **Q10) L231 - how is the DAgger training performed without a human in the loop?**
> > >
> > > We use the same DAgger training technique as the compared baselines [g,h,i]. In the table below, removing schedule sampling (i.e., using ground-truth trajectories at all time) drops the performance. We will add and discuss these results in the supplementary.
> > >
> > > | Training Method | TL ↓ | NE ↓ | OS ↑ | SR ↑ | SPL ↑ |
> > > |:---|:---:|:---:|:---:|:---:|:---:|
> > > | Teacher Forcing | 7.90 | 7.61 | 30.0 | 24.6 | 22.5 |
> > > | Dagger Training | 10.00 | **6.28** | **47.6** | **38.9** | **34.3** |
> > >
> > > > **Q11)  In Tab. 5, classification layer logits are better than semantic map from Tab. 3.**
> > >
> > > Thanks for your comment. In the classification layer variant, the projected features are the logits on 27 categories (i.e., softmax of the predicted scores for each class). In the semantic map variant, the projected features are the image segmentation results (i.e., one-hot encoded the argmax of the scores). We speculate that the logits in the classification layer variant could provide with soft labels to ease the imprecise segmentation results problem caused by the off-the-shelf segmentation model.
> > >
> > > **Reference:**
> > >
> > > [a] MapNet: An Allocentric Spatial Memory for Mapping Environments. CVPR 2018.
> > >
> > > [b] Semantic MapNet: Building Allocentric Semantic Maps and Representations from Egocentric Views. AAAI 2021.
> > >
> > > [c] Cognitive Mapping and Planning for Visual Navigation. CVPR 2017.
> > >
> > > [d] Geometry-Aware Recurrent Neural Networks for Active Visual Recognition. NeurIPS 2018.
> > >
> > > [e] Cross-modal Map Learning for Vision and Language Navigation. CVPR 2022.
> > >
> > > [f] Occupancy Anticipation for Efficient Exploration and Navigation. ECCV 2020.
> > >
> > > [g] Beyond the Nav-Graph: Vision-and-Language Navigation in Continuous Environments. ECCV 2020.
> > >
> > > [h] Language-Aligned Waypoint (LAW) Supervision for Vision-and-Language Navigation in Continuous Environments. EMNLP 2021.
> > >
> > > [i] Bridging the Gap Between Learning in Discrete and Continuous Environments for Vision-and-Language Navigation. CVPR 2022.

---

> > > > ### Comment · Reviewer_kB3d · 2022-08-05
> > > > **Rebuttal response from reviewer kB3d**
> > > >
> > > > I thank the authors for their responses and efforts to address reviewer concerns. At a high-level, the authors have addressed majority of my concerns. One particular point remains unaddressed (see below). I will be happy to raise my rating to 6+ if this is also addressed. Please see detailed comments below.
> > > >
> > > > Q1, Q2 - I agree that the goals are different, and the proposed method aims for something different. My question is whether it achieves something different from what prior work does. What I'm looking for:
> > > > * Replace multi-granularity features with Semantic MapNet features (keeping rest of the pipeline fixed). Note that this is different from fine-grained features used in the method. Semantic MapNet trains the map features end-to-end for semantic reconstruction. The proposed method uses pre-trained image segmentation feature and projects them without any learning.
> > > > * Why do this? Semantic MapNet may be implicitly learning the coarse and fine-grained features by performing top-down semantic segmentation. This is akin to how the image segmentation learns fine-grained features through segmentation training.
> > > >
> > > > Q3, Q4 - Thanks for the comparison with [e] as well as object localization performance. The denseness of the trajectory provides better supervision than sparse waypoints and helps identify instruction-relevant objects. This addresses my concern.
> > > >
> > > > Q5 - Agreed that it is reasonable to expect hallucination to be useful. Having this comparison is convincing.
> > > >
> > > > Q7 - Great. Please hint at these statistics in the paper to emphasize the importance of addressing this ambiguity.
> > > >
> > > > Q6, Q8, Q9, Q10 - Yes, these make sense. Please clarify the points in the paper.
> > > >
> > > > Q11 - Interesting. The logits might be a better option for mapping then.
> > > >
> > > > I also appreciate the change from self-supervised to weakly-supervised. The paper reads more naturally with this change. A minor nit: The method can be named WS-MGMap instead is SS-MGMap to reflect the change to weakly-supervised.

---

> > > > > ### Author Response · Authors · 2022-08-05
> > > > > **Response to Reviewer kB3d**
> > > > >
> > > > > Thank you for the followup and comments! We're pleased that our response addresses the majority of your concerns. We will update the response contents in the revised paper and upload the revision later.
> > > > >
> > > > > As for Q1 and Q2, let us make sure we understand your comment correctly.
> > > > > * Do the "Semantic MapNet features", which the Reviewer suggest us replace our multi-granularity map with, refer to the last layer features of the decoder in Semantic MapNet? If yes, we are running such an experiment and will **update and discuss the results once the experiment finishes**.
> > > > >
> > > > >
> > > > > Please don’t hesitate to let us know if there are any additional clarifications or experiments that we can offer, as we would love to convince you of the merits of the paper. We appreciate your suggestions.

---

> > > > > > ### Author Response · Authors · 2022-08-06
> > > > > > **Response to Reviewer kB3d**
> > > > > >
> > > > > > Thank you for the followup and comments! We're pleased that our response addresses the majority of your concerns, and hope to address the remainder now.
> > > > > >
> > > > > > > **Q1, Q2 - Whether the multi-granularity map achieves something different from what prior work does?**
> > > > > >
> > > > > > We conduct an experiment to replace the multi-granularity map with the last layer features of the decoder in Semantic MapNet. We keep all settings the same except for replacing the map. The results below show that using Semantic MapNet features significantly underperforms our method on val-unseen data split.
> > > > > >
> > > > > > | Map Features | TL ↓      | NE ↓     | OS ↑     | SR ↑     | SPL ↑    |
> > > > > > |------------------------------|-----------|----------|----------|----------|----------|
> > > > > > | Semantic MapNet Features     | 10.27     | 6.72     | 42.1     | 33.1     | 29.0     |
> > > > > > | Multi-granularity Map (Ours) | **10.00** | **6.28** | **47.6** | **38.9** | **34.3** |
> > > > > >
> > > > > > We speculate the reason is that **the Semantic MapNet features contain little object low-level features** while **our map contains both low-level and high-level features**. The detailed analysis is as follows.
> > > > > >
> > > > > >
> > > > > > 1. For the Semantic MapNet, its decoder, which consists of five convolutional layers, can be considered as a segmentation model, performing top-down semantic segmentation. Existing network interpretability works [a, b] indicate that low-level features (e.g., color and texture) dominate at lower layers while high-level features dominate higher layers. In this sense, **the last layer of this decoder, as its highest layer, naturally contains few low-level features.**
> > > > > > 2. By comparison, we project the last layer features of a pretrained U-Net to build our fine-grained map. **Because of the skip-connection mechanism of the U-Net, its last layer features are the combination of features from lower layers and higher layers. These naturally contain both low-level and high-level features.** We concatenate this fine-grained map with a predicted semantic map to further incorporate higher-level information to build our multi-granularity map.
> > > > > >
> > > > > >
> > > > > > Based on the above results and analysis, our multi-granularity map 1) provides richer information in different granularity for VLN, and 2) achieves significantly better performance compared with the existing works.
> > > > > >
> > > > > >
> > > > > >
> > > > > > > **Q3-Q11: At a high-level, the authors have addressed the majority of my concerns.**
> > > > > >
> > > > > > We're pleased that our response addresses the majority of your concerns. We have updated the response contents in the revised paper. The revisions are listed as follows:
> > > > > > * We have added ablation studies on the hallucination module and DAgger training in Sections D and E, respectively in the supplemental material. [Q5, Q10]
> > > > > > * We have added quantitative results about object ambiguity in Section 1. Also, we have visualized some ambiguity cases in Section G. [Q7]
> > > > > > * We have clarified some implementation details in the revised paper. [Q6, Q8, Q9]
> > > > > > * We have changed SS-MGMap to WS-MGMap.
> > > > > >
> > > > > > Please don’t hesitate to let us know if there are any additional clarifications or experiments that we can offer. We appreciate your suggestions. Thanks!
> > > > > >
> > > > > >
> > > > > >
> > > > > > **Reference:**
> > > > > >
> > > > > > [a] Shape or Texture: Understanding Discriminative Features in CNNs. ICLR 2021.
> > > > > >
> > > > > > [b] Interpreting Deep Visual Representations via Network Dissection. TPAMI 2018.

---

> > > > > > > ### Comment · Reviewer_kB3d · 2022-08-06
> > > > > > > **Reviewer response**
> > > > > > >
> > > > > > > Yes, this is the exact experiment I'm looking for. Thanks for presenting the results. I'm inclined to agree with the authors' reasoning that the lack of a UNet for Semantic MapNet model limits it from having fine-grained features. It's largely comparable to the semantic map row from table 3. This addresses my final concern. I'm happy to increase my rating to accept. Please stay tuned for my updated review + score in the next 1-2 days.
> > > > > > >
> > > > > > > I have one question that's more of a curiosity than a concern (so it won't affect my rating or review negatively). Depending on time constraints, the authors can feel free to not respond in detail to this. Will using a UNet encoder-decoder instead of 5-layer decoder for Semantic MapNet achieve something similar to MGMap? That is, if we train a Semantic MapNet model with a UNet encoder-decoder for map prediction, and use the final layer features there.

---

> > > > > > > > ### Author Response · Authors · 2022-08-08
> > > > > > > > **Thank you!**
> > > > > > > >
> > > > > > > > Thank you for the followup and for reconsidering your score! We’re pleased that our response addresses your concerns.
> > > > > > > >
> > > > > > > > As for the Semantic MapNet variant that takes a UNet as a decoder, our MGMap is still different from this variant in the following two aspects. 1) Our MGMap explicitly incorporates a predicted global semantic map, which contains higher-level information compared with implicit features. Also, it allows agents to hallucinate the environment out of view range. Experimental results in Table 3 of the paper and in Q5 (ablation on hallucination) of the original response have proved its effectiveness. 2) The decoder of Semantic MapNet attempts to learn implicit multi-granularity features in map space (i.e., input and output of the decoder are both in map space). In contrast, we extract a fine-grained map directly from original RGB images, which may contain more detailed information. We are not clear about the exact effect brought by this difference yet and it would be interesting to conduct further study in further works.
> > > > > > > >
> > > > > > > > Overall, we believe that introducing MGMap for VLN and exploring weakly-supervised signals for localizing instruction objects in the map is a promising and intriguing direction.

---

> > > > > > > > ### Author Response · Authors · 2022-08-09
> > > > > > > > **Looking Forward to the Updated Review and Rating**
> > > > > > > >
> > > > > > > > Thanks for your valuable reviews. We’re pleased that our response addresses your concerns and the Reviewer is happy to increase the rating to accept.
> > > > > > > >
> > > > > > > > As the rebuttal phase is coming to the end in 10 hours, please don’t hesitate to let us know if there are any additional clarifications or experiments that we can offer. We are really looking forward to your updated review and rating. Thanks!

---

### Official Review · Reviewer_2vTe · 2022-07-12

**Rating:** 8
**Confidence:** 4
**Soundness:** 4 excellent
**Presentation:** 3 good
**Contribution:** 3 good

**Summary:**

This paper presents a map-based method for Vision-and-Language Navigation (VLN). The approach builds a semantic map which includes object categories and properties such as color, texture and shape. The semantic map is used to localize the objects and their properties mentioned in the instruction. The localization, map and instruction is used to predict waypoints for a local policy. Experiments onVision-and-Language Navigation in Continuous Environments (VLN-CE) benchmark show competitive results.

**Questions:**

- I do not understand why the authors use the term "Multi-granularity" for describing maps representing objects and their properties. It seems a bit misleading to me because the granularity of the map for both object and properties is the same (denoted by m L143 and L154). What parts of the map have multiple granularities?
- A potential benefit of a modular map-based approach which is trained offline vs end-to-end RL approaches is the sample efficiency as shown in https://arxiv.org/abs/2201.10029. Did you observe such benefits for the proposed approach as compared to the baselines?

**Limitations:**

Yes.

**Strengths And Weaknesses:**

Strengths:
- The paper presents of novel application of map-based navigation techniques to VLN-CE. The use of maps encoding both object categories and properties for navigation is also novel to the best of my knowledge.
- The use of ground truth trajectories for language-to-map grounding is innovative. It allows the authors to have a localization objective without requiring map annotations.
- The ablation experiments show the importance of both kinds of map.

Weaknesses:
- There is a published method (CWP-CMA) which performs better than the proposed approach on the public VLN-CE benchmark. The benefit of the proposed approach is that it doesn't require the agent to turn-around to get a panorama between waypoints. However, it is unclear which approach would be better given access to a panoramic camera.

---

> ### Author Response · Authors · 2022-08-02
> **Response to Reviewer 2vTe**
>
> We thank you for your time and valuable comments. Below we answer the main concerns raised in the review and would be happy to provide further clarification if suitable.
>
> > **Q1) There is a published method (CWP-CMA) that performs better than the proposed approach on the public VLN-CE benchmark. The benefit of the proposed approach is that it doesn't require the agent to turn-around to get a panorama between waypoints. However, it is unclear which approach would be better given access to a panoramic camera.**
>
> In this paper, we strictly follow the standard setting of VLN-CE [a] to use a camera with limited field of view.
>
> To evaluate the performance of our method under a panoramic setting, we encode 12 views in panoramic RGB images using the same visual encoder and project their features to a top-down map. We report the experimental results on val-unseen data split as below:
>
> | Method | TL ↓ | NE ↓ | OS ↑ | SR ↑ | SPL ↑ |
> |:---|:---:|:---:|:---:|:---:|:---:|
> | CWP-CMA [b] (concurrent) | 10.90 | 6.20 | 52.0 | 41.0 | 36.0 |
> | Ours w/ pano | **10.58** | **6.19** | 50.2 | **41.9** | **37.4** |
>
> From the experimental results, our method performs competitively with the concurrent work CWP-CMA [b]. We believe our performance could be further improved if we carefully design how to use panoramic images. Also, exploiting rich environment information encoded by our multi-granularity map to enhance the candidate waypoint predictor proposed in CWP-CMA is an interesting research direction. We leave this for future works.
>
> > **Q2) It seems a bit misleading to me because the granularity of the map for both objects and properties is the same. What parts of the map have multiple granularities?**
>
> Sorry for the confusion. The term "multi-granularity" in our paper means that our map contains different types of features instead of different map resolutions. Specifically, our multi-granularity map consists of a fine-grained map and a semantic map. The fine-grained map represents low-level detail information, such as texture, and color. The predicted semantic map reveals the categories information of common seen objects. This term is also used in some existing works [d, e] to represent the similar meaning. We will make it more clear in the revised paper.
>
> > **Q3) A potential benefit of a modular map-based approach which is trained offline vs end-to-end RL approaches is the sample efficiency. Did you observe such benefits for the proposed approach as compared to the baselines?**
>
> Yes, we have also observed that our method is much more sample efficient than end-to-end RL approaches. Specifically, a common end-to-end RL approach WPN [c] is trained using around 200M samples (note: we consider one step as one training sample). By comparison, our model is trained using only 13M training samples and achieves better performance on val-unseen data split as shown below:
>
> | Method | TL ↓  | NE ↓    | OS ↑ | SR ↑  | SPL ↑ | # Training Samples↓   |
> |:---|:---:|:---:|:---:|:---:|:---:|:---:|
> | WPN [c] | 7.62 | 6.31 | 40.0 | 36.0 | 34.0 | 200M |
> | Ours | 10.89 | **6.2** | **47.6** | **38.9** | **34.3** | **13M** |
>
> We speculate that end-to-end RL approaches require large amounts of data to learn all navigation components like mapping, planning, and control in a single network.
>
> **Reference:**
>
> [a] Beyond the Nav-Graph: Vision-and-Language Navigation in Continuous Environments. ECCV 2020.
>
> [b] Bridging the Gap Between Learning in Discrete and Continuous Environments for Vision-and-Language Navigation. CVPR 2022.
>
> [c] Waypoint Models for Instruction-guided Navigation in Continuous Environments. ICCV 2021.
>
> [d] Multiple Granularity Descriptors for Fine-Grained Categorization. ICCV 2015.
>
> [e] Multi-granularity for knowledge distillation. Image and Vision Computing 2022.

---

> > ### Comment · Reviewer_2vTe · 2022-08-09
> > **Adequate response**
> >
> > Thanks for answering the questions and the additional analysis. I am increasing my rating to 8.

---

> > > ### Author Response · Authors · 2022-08-10
> > > **Thank you!**
> > >
> > > Thanks for your valuable comments!

---

### Author Response · Authors · 2022-08-02
**General Response to All Reviewers and ACs**

We sincerely appreciate all reviewers’ time and efforts in reviewing our paper and for the constructive feedback. In addition to the response to specific reviewers, here we would like to 1) thank reviewers for their acknowledgment of our work, 2) summarize our contributions 3) highlight the new results added during the rebuttal and 4) higilight the revision in the revised paper:

**1) We are glad that the reviewers appreciate and recognize our contributions.**

- The paper presents of novel application of map-based navigation techniques [2vTe]
- The proposed method is well motivated [kB3d,eYGK,RkNa]
- The experimental results show good improvements over comparable methods [kB3d,eYGK,RkNa]
- The paper performed a thorough set of experiments [2vTe,kB3d,eYGK,RkNa]
- The paper writing is good and it was easy to read [kB3d]



**2) We summarize our contributions as follows.**

We emphasize that we do not claim our main contribution is how to build a fine-grained map or a semantic map. As discussed in the related works and pointed out by Reviewer RkNa, there are existing works focusing on building a fine-grained map from RGB features or a semantic map from semantic segmentation prediction.

However, as shown in Tab. 3 and 4 in the paper, directly using one of these maps achieves inferior performance. The underlying reason is that individually using these maps without specific supervision can not represent diverse objects with different attributes in the VLN instruction and environment. How to learn a map representation, which is representative for diverse objects, for VLN has never been investigated or solved in previous works. Our work aims to answer the questions that how to provide sufficient information and supervision for learning such a map representation for VLN task.

To summarize, our contributions are summarized as followed.

- We are the **first** to **introduce multi-granularity knowledge**, which contains both fine-grained details and semantic information of objects, in a map format to build a multi-granularity map for VLN task.
- We propose a weakly-supervised object localization auxiliary task to provide supervision for the agent to **learn to represent diverse instruction-relevant objects** in the map representation.
- Experimental results on VLN-CE benchmark show our method achieves state-of-the-art results, improving navigation success rate by 4.0% and 4.6% in seen and unseen environments, respectively.

**3) In this rebuttal, we have added more supporting results following the reviewers’ suggestions.**

- Performance of our method under panoramic setting [2vTe]
- Comparison of sample efficiency between WPN and our method on training samples [2vTe]
- Ablation results of different heatmap types for the auxiliary task [kB3d]
- Ablation results of hallucinating maps beyond the field of view [kB3d]
- Evaluation on RxR-Habitat [eYGK]
- Performance of our method without semantic annotation [RkNa]

**4) We make the following modifications in our revision to address reviewers' questions. (highlighted in magenta).**


- We add discussion of prior works that build a map in high-dimensional spaces in the related work section [kB3d, RkNa]
- We remove the motivation of human behavior in the introduction section [eYGK]
- We add more implementation details. [2vTe, kB3d ,RkNa]
- We add more discussion about limitations and future work [RkNa]
- We replace the "self-supervised" description with "weakly-supervised" [eYGK]

---

### Meta-Review · Area_Chair_EMqB · 2022-08-26

**Recommendation:** Accept
**Confidence:** Certain

**Metareview:**

The paper received all positive reviews (3x accept ratings, 1x strong accept rating). The meta-reviewer agrees with the reviewers' assessment of the paper.

**Award:**

No

---

### Decision · Program_Chairs · 2022-09-14

Accept